



# Optimal parameters for the ocean's nutrient, carbon, and oxygen cycles compensate for circulation biases but replumb the biological pump

Benoît Pasquier[1], Mark Holzer[1], Matthew A. Chamberlain[2], Richard J. Matear[2], Nathaniel L. Bindoff[3], and François W. Primeau[4]

[1]School of Mathematics and Statistics, University of New South Wales, Sydney, NSW, Australia
[2]Commonwealth Scientific and Industrial Research Organisation, Hobart, TAS, Australia
[3]Institute for Marine and Antarctic Studies, University of Tasmania, Hobart, TAS, Australia
[4]Department of Earth System Science, University of California, Irvine, CA, USA

**Correspondence:** Benoît Pasquier (b.pasquier@unsw.edu.au)

**Abstract.** Accurate predictive modelling of the ocean's global carbon and oxygen cycles is challenging because of uncertainties in both biogeochemistry and ocean circulation. Advances over the last decade have made parameter optimization feasible, allowing models to better match observed biogeochemical fields. However, does fitting a biogeochemical model to observed tracers using a circulation with known biases robustly capture the inner workings of the biological pump? Here we embed a

mechanistic model of the ocean's coupled nutrient, carbon, and oxygen cycles into two circulations for the current climate. To assess the effects of biases, one circulation (ACCESS-M) is derived from a climate model and the other from data assimilation of observations (OCIM2). We find that parameter optimization compensates for circulation biases at the expense of altering how the biological pump operates. Tracer observations constrain pump strength and regenerated inventories for both circulations, but ACCESS-M export production optimizes to twice that of OCIM2 to compensate for ACCESS-M having lower sequestra-

tion efficiencies driven by less efficient particle transfer and shorter residence times. Idealized simulations forcing complete Southern Ocean nutrient utilization show that the response of the optimized system is sensitive to the embedding circulation. In ACCESS-M, Southern Ocean nutrient and DIC trapping is partially short-circuited by unrealistically deep mixed layers. For both circulations, intense Southern Ocean production deoxygenates Southern-Ocean-sourced deep waters, muting the imprint of circulation biases on oxygen. Our findings highlight that the biological pump's plumbing needs careful assessment to predict

the biogeochemical response to environmental changes, even when optimally matching observations.

## 1   Introduction

The ocean's nutrient, carbon, and oxygen cycles are of central importance for climate and the fertility of the ocean. The cycling rates and patterns are shaped by the subtle interplay between the ocean circulation and the generation, transport, and respiration of organic matter (the biological pump) as well as by air–sea gas exchange. Building robust predictive models of

the ocean's biological pump poses a formidable challenge because of the myriad of biogeochemical processes that must be parameterized. Current prognostic earth-system models are computationally expensive, which prohibits systematic parameter



space exploration. Relatedly, the long timescales of the global ocean circulation make it expensive to spin up earth-system models by brute-force time stepping and differences among models have been shown to reflect differences in spin-up strategy (Séférian et al., 2016), compounding the parametric uncertainties. As a result of these computational challenges, simulations

with earth-system models have made widely varying predictions of ocean biogeochemistry (Bopp et al., 2013; Cocco et al., 2013; Henson et al., 2022).

To reduce the parametric uncertainties, biogeochemical parameters can be objectively determined by minimizing the quadratic mismatch between model-predicted and observed tracer distributions. Systematic parameter optimization is made possible by embedding the biogeochemical model in climatological steady flow and efficiently solving the generally nonlinear equations

of the system for steady state directly using Newton-type implicit solvers (e.g., Kwon and Primeau, 2006). This approach exploits the matrix representation of the discretized advective–diffusive flux-divergence operator (the "transport matrix", e.g., Khatiwala et al., 2005; Primeau, 2005; Chamberlain et al., 2019) and has been applied to a number of biogeochemical cycles embedded in data-assimilated ocean circulation models (e.g., Primeau et al., 2013; DeVries, 2014; Teng et al., 2014; Holzer et al., 2014; Pasquier and Holzer, 2017; DeVries and Weber, 2017; Wang et al., 2019, to cite a few).

When the circulation is data-assimilated to provide a realistic representation of the ocean's advective–eddy-diffusive transport, optimizing biogeochemical parameters is a natural strategy for obtaining robust representations of the ocean's biogeochemical cycles. However, Kriest and Oschlies (2015) demonstrated that optimal parameters for the Model of Oceanic Pelagic Stoichiometry (MOPS; Kriest and Oschlies, 2015) differ depending on the circulation model that is used. This raises the questions: If the circulation is taken from a climate-model simulation with known biases in the ocean's physical state, do optimized

biogeochemical parameters still provide a reliable estimate of the ocean's biogeochemistry, and are the simulated responses of the system to either biogeochemical or physical perturbations robust?

To answer these questions, we develop a relatively simple model (dubbed PCO2 here) of the coupled phosphorous, carbon, and oxygen cycles and contrast the properties of its biological pump and its response to perturbations depending on whether the model is optimized for a data-assimilated circulation or for a climate-model-derived circulation. The PCO2 model was

constructed with a particular focus on capturing the coupling between oxygen and organic particle respiration. We use a mechanistic formulation of nutrient uptake rather than an observation-based parameterization so that biological production can fully respond to the embedding circulation (and also to make PCO2 suitable for exploring climate-change scenarios in future studies). PCO2 improves on the well-established OCMIP approach (Najjar et al., 1992) in that the functional form of the particle-flux divergence is not specified at the outset to be either a power law or exponential. Instead, we explicitly

model sinking biogenic particles and let them react with the ambient oxygen in a simple temperature-dependent model of microbial respiration (DeVries and Weber, 2017; Laufkötter et al., 2017) to determine the flux divergence and respiration rates mechanistically.

We focus on a decadal-mean circulation derived from the ACCESS-1.3 climate model (Chamberlain et al., 2019; Holzer et al., 2020) for the 1990s. To assess PCO2's biogeochemistry and biological pump when embedded and optimized in this

climate-model circulation, we compare to PCO2 embedded and optimized in a data-assimilated ocean circulation (OCIM2, DeVries and Holzer, 2019). OCIM2 has been optimized so that its transport produces tracer fields that are as close as possible





to observations. Thus, OCIM2 provides a realistic reference point with minimal circulation biases. To explore the effect of optimized biogeochemical parameters on the response to biogeochemical perturbations, we consider idealized perturbations in which biological production in the Southern Ocean is intensified to cause a nearly complete nutrient drawdown. Similar

perturbations have been used previously to quantify the Southern Ocean's key role in supplying the rest of the ocean with preformed nutrients (e.g., Sarmiento et al., 2004; Marinov et al., 2006; Holzer and Primeau, 2013) and to illustrate Southern Ocean nutrient trapping (Primeau et al., 2013). Here, we establish how the response to Southern Ocean nutrient drawdown differs depending on whether biogeochemistry is optimized for the OCIM2 or ACCESS circulations.

We find that the biogeochemical model can be optimized to fit the observed phosphate, dissolved inorganic carbon, oxygen,

and total alkalinity distributions with reasonable fidelity for both data-assimilated and climate-model derived circulations. However, we find that the biological pump operates very differently for the OCIM2 and ACCESS-M circulations, largely because of differences in the sequestration time of regenerated organic matter. The differences in the biological pump in turn produce significant differences in the response of the system to imposed Southern Ocean nutrient drawdown.

## 2   Methods

### 2.1   Biogeochemical Model

We model the ocean's coupled phosphorus, carbon, and oxygen cycles using mechanistic representations of nutrient uptake, particle transport, and respiration. Depending on whether oxygen is prescribed by observations or explicitly modelled, we refer to the model as the PC or PCO2 model, respectively.

Biological production is approximated as requiring only phosphate as a nutrient and the production of organic carbon is

keyed to phosphate uptake. For simplicity, DOP is deemed to be not bioavailable, although it has been shown to be utilized in oligotrophic regions (Letscher et al., 2016). Nitrate, silicic acid, and iron limitations are not explicitly modelled either, although we do parameterize the effect of denitrification on respiration as described below. We justify our approximations *a posteriori* by the fidelity of the modelled fields to observations.

We model 4 distinct phosphorus pools: dissolved inorganic phosphorus (DIP, which is phosphate, $PO_4$), semi-labile dissolved

organic phosphorus (DOP), and fast and slowly sinking particulate organic phosphorus ($POP_f$ and $POP_s$). The steady-state equations for these tracers are

$$
\begin{aligned}
\mathcal{T}\,[\mathrm{DIP}] &= & -U_\mathrm{P} & & +R_\mathrm{DOP}+R_\mathrm{POP} & & +J_\mathrm{DIP}^\mathrm{geo} \\
\mathcal{T}\,[\mathrm{DOP}] &= & \sigma U_\mathrm{P} & & -R_\mathrm{DOP} & +D_\mathrm{POP} \\
\mathcal{S}_\mathrm{f}\,[\mathrm{POP_f}] &= & \sigma_\mathrm{f}\,(1-\sigma)\,U_\mathrm{P} & & -R_\mathrm{POP_f}-D_\mathrm{POP_f} \\
\mathcal{S}_\mathrm{s}\,[\mathrm{POP_s}] &= & (1-\sigma_\mathrm{f})\,(1-\sigma)\,U_\mathrm{P} & & -R_\mathrm{POP_s}-D_\mathrm{POP_s}
\end{aligned}
\tag{1}
$$

with transport terms (circulation or gravitational settling) on the left and sources and sinks on the right. Specifically, $\mathcal{T}\,[\mathrm{X}] = \nabla \cdot (\boldsymbol{u}\,[\mathrm{X}]) - \nabla \cdot (\mathbf{K}\,\nabla[\mathrm{X}])$ is the flux divergence of dissolved tracer X due to advection (velocity $\boldsymbol{u}$) and eddy diffusion (dif-





fusivity tensor $\mathbf{K}$). Similarly, $\mathcal{S}_k[\mathrm{POP}_k] = \partial_z(w_k[\mathrm{POP}_k])$ is the divergence of the flux of the $\mathrm{POP}_k$ tracer with sinking speed $w_k$ (where $k = \mathrm{f}$ or $k = \mathrm{s}$). A fraction $\sigma$ of the phosphorus uptake $U_\mathrm{P}$ is allocated to DOP, a fraction $\sigma_\mathrm{f}(1-\sigma)$ is allocated to $\mathrm{POP}_\mathrm{f}$, and the remainder is allocated to $\mathrm{POP}_\mathrm{s}$, all of which are remineralized back into the DIP pool (through $R_\mathrm{DOP}$ and $R_\mathrm{POP} = R_{\mathrm{POP}_\mathrm{f}} + R_{\mathrm{POP}_\mathrm{s}}$). The global phosphate inventory is prescribed by weakly restoring DIP to the global observed mean $\overline{[\mathrm{DIP}]} = 2.17\,\mu\mathrm{M}$ via $J_\mathrm{DIP}^\mathrm{geo} = (\overline{[\mathrm{DIP}]} - [\mathrm{DIP}])/\tau_\mathrm{geo}$ with "geological" timescale $\tau_\mathrm{geo} = 1\,\mathrm{Myr}$. Remineralization

of organic phosphorus and respiration of organic carbon are modelled as having the same specific (i.e., per molecule) rates so that remineralization preserves the C:P ratio of organic-matter production (discussed below). We now briefly describe how the remineralization/respiration rates are modelled; details of the phosphorus uptake rate per unit volume $U_\mathrm{P}$ are provided in Appendix A1.

The remineralization of particulate organic matter (POM; either POP or particulate organic carbon, POC) is known to have

relatively simple dependencies on oxygen and temperature that are parameterized explicitly following previous work (e.g., Laufkötter et al., 2017; DeVries and Weber, 2017; Dinauer et al., 2022):

$$R_{\mathrm{POM}_k} = \gamma_k\, q_{10}^{\frac{T-T_\mathrm{ref}}{10\,\mathrm{K}}} \frac{\max([\mathrm{O}_2], [\mathrm{O}_2^\mathrm{lim}])}{\max([\mathrm{O}_2], [\mathrm{O}_2^\mathrm{lim}]) + K_{\mathrm{O}_2}}[\mathrm{POM}_k], \tag{2}$$

where $T_\mathrm{ref} = 20\,^\circ\mathrm{C}$. Note, however, that Eq. (2) differs from previous parameterizations in that we implicitly include the effect of microbes switching to nitrate for organic matter oxidization (denitrification) by disallowing respiration rates to decline in

anoxic waters below $[\mathrm{O}_2^\mathrm{lim}] = 5\,\mu\mathrm{M}$. To limit unrealistic POM accumulation in the bottom grid boxes under anoxic conditions, a small fraction of POM is dissolved into DOM at rate $D_\mathrm{POM} = [\mathrm{POM}]/\tau_\mathrm{POM}$ with $\tau_\mathrm{POM} = 1\,\mathrm{yr}$. Remineralization of dissolved organic matter (DOM; either DOP or dissolved organic carbon, DOC) is approximated as $R_\mathrm{DOM} = [\mathrm{DOM}]/\tau_\mathrm{DOM}$ with a simple globally uniform time scale $\tau_\mathrm{DOM} = 2\,\mathrm{yr}$.

The steady-state equations for dissolved inorganic carbon (DIC), DOC, fast and slow POC ($\mathrm{POC}_\mathrm{f}$ and $\mathrm{POC}_\mathrm{s}$), and particulate

inorganic carbon (PIC, which is $\mathrm{CaCO}_3$) are

$$
\begin{aligned}
\mathcal{T}[\mathrm{DIC}] &= -(1 + r_\mathrm{PIC}(1-\sigma))U_\mathrm{C} + R_\mathrm{DOC} + R_\mathrm{POC} && + D_\mathrm{PIC} + J_\mathrm{DIC}^\mathrm{atm} \\
\mathcal{T}[\mathrm{DOC}] &= \sigma U_\mathrm{C} && - R_\mathrm{DOC} && + D_\mathrm{POC} \\
\mathcal{S}_\mathrm{f}[\mathrm{POC}_\mathrm{f}] &= \sigma_\mathrm{f}(1-\sigma)U_\mathrm{C} && - R_\mathrm{POCf} - D_\mathrm{POCf} \\
\mathcal{S}_\mathrm{s}[\mathrm{POC}_\mathrm{s}] &= (1-\sigma_\mathrm{f})(1-\sigma)U_\mathrm{C} && - R_\mathrm{POCs} - D_\mathrm{POCs} \\
\mathcal{S}_\mathrm{PIC}[\mathrm{PIC}] &= r_\mathrm{PIC}(1-\sigma)U_\mathrm{C} && - D_\mathrm{PIC}
\end{aligned}
\tag{3}
$$

where $\mathcal{S}_\mathrm{PIC}[\mathrm{PIC}] = \partial_z(w_\mathrm{PIC}[\mathrm{PIC}])$ is the flux divergence of PIC sinking at speed $w_\mathrm{PIC}$. The uptake rate of carbon per unit volume $U_\mathrm{C} = r_\mathrm{C:P}U_\mathrm{P}$ is keyed to phosphate uptake using the stoichiometric C:P ratio $r_\mathrm{C:P}$, parameterized here in terms of $[\mathrm{DIP}]$ (Galbraith and Martiny, 2015, see also Appendix A2). Uptake of DIC results in the production of DOC, $\mathrm{POC}_\mathrm{f}$, and

$\mathrm{POC}_\mathrm{s}$ in the same proportions as the corresponding phosphorus tracers (determined by $\sigma$ and $\sigma_\mathrm{f}$; see Eq. (1)). For OCIM2, we account for the effect of precipitation and evaporation on DIC with "virtual fluxes" (Murnane et al., 1999) as described in




the OCMIP protocol (Najjar and Orr, 1999). The ACCESS-M matrix captures the flux divergence due to water exchange with the atmosphere directly. The carbonate pump is keyed to the soft-tissue pump via the rain ratio $r_{\mathrm{PIC}} = \mathrm{PIC:POC}$, and PIC dissolution is parameterized as $D_{\mathrm{PIC}} = [\mathrm{PIC}]/\tau_{\mathrm{PIC}}$ with $\tau_{\mathrm{PIC}} = 1\,\mathrm{d}$.

In Equation (3), $J_{\mathrm{DIC}}^{\mathrm{atm}}$ is the DIC source/sink term due to air–sea $\mathrm{CO_2}$ exchange, parameterized in terms of surface winds and sea-ice fractions using the formulation of Wanninkhof (1992) with prescribed preindustrial atmospheric $\mathrm{pCO_2} = 278\,\mu\mathrm{atm}$. (We optimize our model for preindustrial conditions, assuming negligible changes in circulation since preindustrial times.) For OCIM2, we use National Centers for Environmental Prediction (NCEP) reanalyses for the ice fraction and 6-hourly surface winds, while for ACCESS-M we use the corresponding quantities as simulated by the ACCESS climate model. From these
winds, 6-hourly piston velocities are computed (to capture gustiness) that are combined with ice-fraction and Schmidt-number information and time averaged to form an annual-mean climatology of the gas exchange coefficients. The effective partial pressure of $\mathrm{CO_2}$ in seawater needed for air–sea $\mathrm{CO_2}$ exchange is calculated from the equilibrium carbonate chemistry using the MATLAB CO2SYS function (Lewis and Wallace, 1998; van Heuven et al., 2011).

The sinking speeds of the biogenic particles ($w_{\mathrm{s}}$ and $w_{\mathrm{f}}$ for POM and $w_{\mathrm{PIC}}$ for PIC) are constructed from globally uniform
sinking speeds ($w_{\mathrm{s}}^{*}$, $w_{\mathrm{f}}^{*}$, and $w_{\mathrm{PIC}}^{*}$) that are multiplied with a dimensionless in-situ viscosity factor $\alpha_{\mu}$ to account for slower terminal velocities in colder (and to a lesser degree in more saline) waters. $\alpha_{\mu}$ depends on seawater viscosity and on the density difference between POM and ambient seawater (Taucher et al., 2014, Appendix A3).

The concentration of total alkalinity (TA) obeys (Murnane et al., 1999)

$$\mathcal{T}[\mathrm{TA}] = 2\left(D_{\mathrm{PIC}} - r_{\mathrm{PIC}}(1-\sigma)U_{\mathrm{C}}\right) - 21.8\left(R_{\mathrm{DOP}} + R_{\mathrm{POP_f}} + R_{\mathrm{POP_s}} - U_{\mathrm{P}}\right) + J_{\mathrm{TA}}^{\mathrm{geo}}. \tag{4}$$

The first term represents sources and sinks of TA due to the cycling of carbonate ($\mathrm{TA:C} = 2$) and the second term contains the contributions from nitrate ($\mathrm{TA:P} = 16$), phosphate ($\mathrm{TA:P} = 1$), and sulphate ($\mathrm{TA:P} = 2 \times 2.4$) cycling (stoichiometric ratios from Wolf-Gladrow et al., 2007). We approximate the TA inventory as being conserved and hence set the global mean TA value via weak restoring to $\overline{[\mathrm{TA}]} = 2420\,\mu\mathrm{M}$ via the $J_{\mathrm{TA}}^{\mathrm{geo}}$ term, analogous to what we do for phosphate. For OCIM2, "virtual fluxes" are again used to account for concentration and dilution of TA by evaporation and precipitation.

The concentration of dissolved oxygen $[\mathrm{O_2}]$ is set by air–sea gas exchange, organic-matter respiration, and phytoplankton photosynthesis. In steady-state $[\mathrm{O_2}]$ obeys

$$\mathcal{T}[\mathrm{O_2}] = r_{\mathrm{O_2:C}}\left[U_{\mathrm{C}} - (R_{\mathrm{DOC}} + R_{\mathrm{POC_f}} + R_{\mathrm{POC_s}})\,\Theta([\mathrm{O_2}] - [\mathrm{O_2^{lim}}])\right] + J_{\mathrm{O_2}}^{\mathrm{atm}}, \tag{5}$$

where $r_{\mathrm{O_2:C}}$ is the $\mathrm{O_2:C}$ stoichiometric ratio of organic matter modelled as globally uniform. The Heaviside (step) function $\Theta$ switches oxygen consumption off when $[\mathrm{O_2}]$ falls below $[\mathrm{O_2^{lim}}] = 5\,\mu\mathrm{M}$, consistent with the parameterization of anaerobic
POM respiration in Eq. (2). The air–sea exchange rate $J_{\mathrm{O_2}}^{\mathrm{atm}}$ for oxygen is parameterized similarly to that for $\mathrm{CO_2}$ (Wanninkhof, 1992) using the coefficients for oxygen solubility and Schmidt number as tabulated by Wanninkhof (2014).

## 2.2 Steady-state ocean circulation models

The nonlinear coupled partial differential equations (1), (3), (4), and (5) are discretized on the model grid and the three-dimensional tracer fields are organized into column vectors. Linear operators such as $\mathcal{T}$ and $\mathcal{S}$ then become sparse matrices,





usually referred to as transport matrices, especially when referring to advection/diffusion. The discretized steady-state equations are coupled nonlinear algebraic equations that are solved using Newton's method, requiring order 10 iterations (Appendix B).

### 2.2.1   OCIM2

The Ocean Circulation Inverse Model version 2 (OCIM2; DeVries, 2014; DeVries and Holzer, 2019) provides a data-assimilated
advection–eddy-diffusion transport matrix. The OCIM2 data assimilation uses the ventilation tracers CFC-11, CFC-12, radiocarbon, and $^3$He, in addition to sea-level height and air–sea heat and fresh-water fluxes. OCIM2 has a horizontal resolution of $2° \times 2°$ and 24 levels with layer thicknesses that increase with depth. The OCIM2 transport operator is an $N \times N$ sparse matrix with $N \approx 2 \times 10^5$ and $3 \times 10^6$ nonzero elements. OCIM2 arguably provides the most realistic estimate of the ocean's climatological steady-state transport and is thus a natural reference against which to assess biases in climate-model-derived
transport for the current state of the ocean.

### 2.2.2   ACCESS-M

As a climate-model based estimate of the ocean's advection–diffusion operator, we use a slightly modified version of the "preferred" ACCESS1.3 transport matrix of Chamberlain et al. (2019). This matrix was built from the decadal mean volume fluxes (resolved plus parameterized) for the 1990s from the ACCESS1.3 "historical" runs (Bi et al., 2013a) with nominal $1° \times 1°$
horizontal resolution (finer in latitude near the equator) and 50 vertical levels. This ACCESS1.3 matrix has a size of $N \times N$ with $N \approx 2.7 \times 10^6$ and $1.8 \times 10^7$ nonzeros entries, which is an order of magnitude larger than the OCIM2 matrix.

    The tripolar grid of ACCESS1.3 results in a more complex sparsity pattern that slows the factorization of the Jacobian in Newton's method (Appendix B1). We therefore coarse-grain the ACCESS1.3 matrix by lumping together 2×2 nearest horizontal neighbors (similar to the "lump-and-spray" approach of Bardin et al. (2014)), which results in about 16× faster factorization.
This coarse-graining reduces the maximum ideal mean age in the Pacific, but we compensate by reducing the interior background diffusivity from $0.3\,\mathrm{cm^2\,s^{-1}}$ to $0.1\,\mathrm{cm^2\,s^{-1}}$ to match OCIM2 and to retain the original ideal mean age. We refer to the resulting matrix/circulation model as ACCESS-M, which is of size $N \times N$ with $N \approx 7 \times 10^5$ and $5 \times 10^6$ nonzero entries. We emphasize that high resolution is not important for transport matrices built from the output of an ocean model because the model's volume fluxes already contain the mean effects of processes resolved (and parameterized) at higher resolution in the
parent circulation model.

    The most important difference between ACCESS-M and OCIM2 for simulating biogeochemistry stems from differences in how the mixed layer is modelled. Both matrix models use mean annual maximum mixed-layer depth (MLD). However, while OCIM2 specifies MLD from observational analyses (de Boyer Montégut et al., 2004), ACCESS-M uses the MLD of the parent ocean model. Overall, the ACCESS-M MLD is deeper than observed (roughly 1.5–3 times in the subtropical gyres) and has
important unrealistic features. In the Weddell and Ross Seas, the ACCESS-M MLD reaches all the way to the sea floor and the deep winter mixed layers of the North Atlantic and Nordic Seas occupy a much larger area than observed (Appendix C). The deep Southern Ocean mixed layers are due to unrealistic open-ocean convection (Bi et al., 2013a; Heuzé et al., 2013)

 

and are a key ACCESS-M feature that imprints on the biological pump and affects its responses to perturbations (see below). Furthermore, ACCESS-M is simply built from time-averaged model volume fluxes, while OCIM2 has a steady transport that is

optimized to yield propagated tracer concentrations that are as close as possible to observations. ACCESS-M therefore inherits documented circulation and thermodynamic biases from the parent ocean model (Marsland et al., 2013; Bi et al., 2013b, 2020).

## 2.3 Data and Parameter Optimization and Tracer Data

We optimize the PCO2 model parameters by minimizing an objective ("cost") function that measures the quadratic mismatch with observed DIP, DIC, $O_2$, and TA and penalizes deviations from a plausible range of values for each parameter. Details on

the objective function and optimization procedure are provided in Appendix B4.

For DIP observations we use gridded annual mean phosphate from the World Ocean Atlas 2018 (Garcia et al., 2019). Gridded $O_2$, DIC, and TA observations are taken from the Global Data Analysis Project (Key et al., 2015; Lauvset et al., 2016, GLODAP v2;). We optimize PCO2 for preindustrial conditions, assumed to be reasonably well represented by the OCIM2 and ACCESS-M circulations and by the observational DIP, TA, and $O_2$ climatologies. For DIC, we subtract an estimate of

anthropogenic DIC as propagated from the reconstructed atmospheric $CO_2$ time history since 1720 using the data-assimilated OCIM2 (as done by Holzer et al., 2021b). The observed tracers are interpolated onto the grid of each circulation model, and grid cells without observations are ignored in the objective function.

## 3 Results

We now focus on the optimized steady state of the PCO2 model and how it differs depending on whether PCO2 is embedded

in the OCIM2 or ACCESS-M circulation. To examine the sensitivity of optimized model parameters to model complexity, we will also consider the PC model, where $O_2$ concentrations are prescribed by observations.

## 3.1 Fidelity to observed fields

To quantify how well the optimized PCO2 model matches observations, we first examine the joint volume-weighted modelled–observed probability density functions (PDFs), which are essentially binned model-versus-observation scatter plots. These are

shown in Fig. 1a–h together with globally averaged depth profiles (Fig. 1j–l) for DIP, $O_2$, DIC, and TA as obtained with either the OCIM2 or ACCESS-M circulations. Overall, there is good agreement (tight clustering of the PDFs around the 1:1 line) with volume-weigthed root-mean-square errors (RMSE) that are around 20–30 % of the observed standard deviation for OCIM2 and around 40–50 % for ACCESS-M.

The optimized OCIM2-embedded PCO2 model compares well to other objectively optimized models of the P, C, and $O_2$

cycles (e.g., Primeau et al., 2013; Pasquier and Holzer, 2016; Holzer, 2022) as quantified by similar RMSEs (PDF panels of Figure 1). However, unlike these other models, PCO2 has interactive oxygen providing mechanistic respiration and reminer-alization. Dissolved oxygen, with its large dynamic range from near zero to about $300\,\mu M$, is the tracer that has the largest mismatch with observations at an RMSE of 34 % of the spatial standard deviation from the global mean. The global mean





**Figure 1.** (a–h) Joint model–observations probability density functions (PDFs) for DIP, $O_2$, DIC, and TA as optimized by the PCO2 model embedded in the OCIM2 (a–d) and ACCESS-M (e–h) circulations. The darker the colors the denser the PDF such that $n\%$ of the data lies outside of the $n$-th percentile contour. The volume-weighted root-mean-square error (RMSE) is indicated in each panel along with its size relative to the (spatial) mean and standard deviation (SD) of the observations. (i–l) Corresponding simulated and observed global-mean vertical profiles. (Because of interpolation, the "observed" profiles depend slightly on the grid used.)

vertical [$O_2$] profile matches the observations above 1000 m but progressively overestimates deeper concentrations, reaching a
high bias of about 15 μM at 4000 m (Fig. 1j).



The optimized ACCESS-M-embedded PCO2 model performs worse for every tracer, with RMSEs that are larger by a factor of 1.4–2.4 (Fig. 1e–h). In contrast to the OCIM2 PCO2 model, the ACCESS-M PCO2 model underestimates oxygen at low concentrations. This could in part be due to ACCESS-M's finer low-latitude resolution, which may allow for more rapid nutrient supply, POM production, and in turn higher oxygen utilization rates. Despite the large local mismatches visible in the joint
PDFs, the horizontally averaged ACCESS-M PCO2 tracer profiles fit the observations reasonably well (Fig. 1i–l), with the exception of $[O_2]$ between 400 m to 1500 m depth where underestimates by up to 30 μM (the largest profile mismatch across models and tracers) indicate that ACCESS-M PCO2 has underoxygenated intermediate waters.

The basin zonal means of Figures 2 and 3 show striking differences between the OCIM2 and ACCESS-M PCO2 oxygen and DIC fields. In the Southern Ocean, ACCESS-M PCO2 strongly overestimates $[O_2]$ (by up to 80 μM) in the same region where
ACCESS-M has unrealistic deep mixing (Fig. C1). This overestimate turns out not to be due to increased preformed oxygen (not shown), which is similar for OCIM2 and ACCESS-M. Instead, the Southern Ocean POC respiration rate is weaker for ACCESS-M, allowing more oxygen to be mixed throughout the water column. (The reduced respiration rate is largely driven by a lower optimal value of $\gamma_s$ (Table 1), with $POC_s$ dominating respiration for ACCESS-M as further discussed in subsection 3.3.2.) In the mid- and low-latitude Atlantic, the OCIM2 PCO2 generally overestimates oxygen especially in the OMZs, while
ACCESS-M PCO2 underestimates oxygen, especially in the thermocline (by up to 80 μM). The underestimates of ACCESS-M PCO2 are consistent with its generally strengthened export production (discussed with Fig. 4 below), producing more organic matter and hence having higher oxygen demand than OCIM2 PCO2. We find that ACCESS-M mode and intermediate waters have a weaker preformed oxygen supply (by order 40 μM, not shown), which also contributes to the large underestimates. In the Pacific, both OCIM2 and ACCESS-M generally underestimate $O_2$ in low latitudes and overestimate it elsewhere, but the
underestimate for ACCESS-M is roughly twice that for OCIM2, again consistent with increased oxygen demand and under-ventilated mode/intermediate waters. In the Indian Ocean, the mismatches are similar in pattern but of larger amplitude for ACCESS-M PCO2.

The zonal-mean $[O_2]$ and $[DIC]$ mismatches in Figures 2 and 3 approximately mirror each other with a Pearson correlation coefficient of about $-0.6$. To the extent that $O_2$ and DIC have realistic air–sea exchange, this anticorrelation is consistent with
higher oxygen corresponding to reduced oxygen utilization and hence reduced DIC production. The details of the mismatch with observations are also influenced by errors in air–sea exchange, but the prominent mirroring of the $O_2$ and DIC mismatches suggests that errors in oxygen utilization are the dominant driver. Regionally in the North Pacific the overall anticorrelation does not hold for ACCESS-M suggesting that other factors play a role there.

When $O_2$ is prescribed from observations (PC model) rather than explicitly simulated, the mismatch improves for most
tracers, despite oxygen not being self consistent. Specifically, we find that relative to PCO2, the RMSEs of the PC model for DIP and DIC improve by 15 and 3.1 % for OCIM2, and by 1.1 and 5.8 % for ACCESS-M. For TA, the mismatch improves by 1.8 % for ACCESS-M but degrades slightly by 0.5 % for OCIM2.



**Figure 2.** (a–c) Basin zonal means of $[O_2]$ from PCO2 embedded in OCIM2 for the Atlantic (a, left), Pacific (b, center), and Indian Ocean (c, right). (d–f) Model-observation difference. (g–l) As (a–f) but for ACCESS-M. Light gray indicates missing observations (see main text for details).

## 3.2 Parameter sensitivity to circulation and model complexity

How sensitive are the values of the optimized biogeochemical parameters to whether we use the OCIM2 or ACCESS-M circulations, and how much do they depend on whether we prescribe the oxygen concentration from observations or simulate $[O_2]$ self-consistently? Recently Kriest et al. (2020) showed that different circulations generally require different parameter values to best match observations. While Kriest et al. (2020) demonstrated this in the context of the Model of Oceanic Pelagic Stoichiometry (MOPS; Kriest and Oschlies, 2015) for 5 circulations, here we address the question for 2 other circulations (OCIM2 and ACCESS) using an entirely different model of biogeochemistry (PCO2). In addition, we investigate how sensitive



**Figure 3.** As Figure 3 but for DIC.

optimized parameter values are to model complexity in the sense of whether oxygen is prescribed (PC model) or computed internally (PCO2 model). The optimized values of the PC and PCO2 parameters for both the OCIM2 and ACCESS-M cases are collected in Table 1.

The overall amplitude $p_{\mathrm{max}}$ and half-saturation constant $K_{\mathrm{DIP}}$ control nutrient uptake and are thus key for P, C, and $O_2$ cycling. While $p_{\mathrm{max}}$ shows sensitivity to both circulation and complexity, it is more sensitive to circulation as quantified by the mean relative standard deviations of $\Delta_{\mathrm{circ}}(p_{\mathrm{max}}) = 64\%$ and $\Delta_{\mathrm{bgc}}(p_{\mathrm{max}}) = 38\%$. (See Appendix E for definitions of $\Delta_{\mathrm{bgc}}$ and $\Delta_{\mathrm{circ}}$.) By contrast, $K_{\mathrm{DIP}}$ is less sensitive to both circulation and complexity, lying in a range of 1.8 to 3.1 µM across models.

The carbonate pump is controlled by the rain ratio $r_{\mathrm{PIC}}$, the fraction $1 - \sigma_{\mathrm{DOM}}$ of production allocated to POM, the sinking-speed parameter $w^*_{\mathrm{PIC}}$, and the PIC dissolution timescale, $\tau_{\mathrm{PIC}}$. These parameters are sensitive to circulation with the OCIM2-



**Table 1.** Parameter values. (Top) Optimized for each model. $\Delta X_{\mathrm{bgc}}$ and $\Delta X_{\mathrm{circ}}$ are the sensitivities to model complexity and circulation (Appendix E). (Bottom) Fixed (non-optimized) parameters.

| Parameter | OCIM2 | | ACCESS-M | | Unit | $\Delta_{\mathrm{bgc}}X$ | $\Delta_{\mathrm{circ}}X$ |
|---|---|---|---|---|---|---|---|
| | PC | PCO2 | PC | PCO2 | | % | % |
| $p_{\max}$ | 4.80 | 12.8 | 19.9 | 23.4 | µM | 38 | 64 |
| $K_{\mathrm{DIP}}$ | 1.81 | 2.86 | 3.00 | 3.14 | µM | 17 | 21 |
| $r_{\mathrm{PIC}}$ | 6.72 | 6.28 | 2.25 | 1.02 | % | 29 | 86 |
| $w^*_{\mathrm{PIC}}$ | 3800 | 3490 | 2010 | 2060 | $\mathrm{m\,d^{-1}}$ | 3.8 | 40 |
| $\sigma$ | 64.8 | 63.9 | 47.1 | 24.2 | % | 23 | 43 |
| $\sigma_{\mathrm{f}}$ | 15.3 | 13.7 | 32.1 | 10.3 | % | 40 | 35 |
| $\gamma_{\mathrm{f}}$ | 0.148 | 0.156 | 0.995 | 1.05 | $\mathrm{d^{-1}}$ | 3.8 | 100 |
| $\gamma_{\mathrm{s}}$ | 0.139 | 0.165 | 0.567 | 0.535 | $\mathrm{d^{-1}}$ | 8.0 | 80 |
| $q_{10}$ | 1.65 | 1.78 | 3.21 | 4.25 | - | 12 | 52 |
| $K_{\mathrm{O_2}}$ | 7.53 | 7.43 | 6.32 | 3.01 | µM | 26 | 36 |
| $r_{\mathrm{O_2:C}}$ | 1.39 | 1.49 | 1.40 | 1.31 | $\mathrm{molO_2\,molC^{-1}}$ | 4.9 | 4.9 |
| $\tau_U$ | | 30 | | | d | | |
| $\kappa_T$ | | 0.063 | | | $\mathrm{K^{-1}}$ | | |
| $K_I$ | | 10 | | | $\mathrm{W\,m^{-2}}$ | | |
| $\tau_{\mathrm{DOM}}$ | | 2 | | | yr | | |
| $\tau_{\mathrm{PIC}}$ | | 1 | | | d | | |
| $\tau_{\mathrm{POM}}$ | | 1 | | | yr | | |
| $w^*_{\mathrm{f}}$ | | 100 | | | $\mathrm{m\,d^{-1}}$ | | |
| $w^*_{\mathrm{s}}$ | | 10 | | | $\mathrm{m\,d^{-1}}$ | | |
| $T_{\mathrm{ref}}$ | | 20 | | | °C | | |
| $[\mathrm{O_2^{lim}}]$ | | 5 | | | µM | | |

embedded PCO2 exporting more PIC to greater depth: For OCIM2, $r_{\mathrm{PIC}}(1-\sigma_{\mathrm{DOM}}) = 2.3\,\%$ and $w_{\mathrm{PIC}}/\tau_{\mathrm{PIC}} = 3500\,\mathrm{m}$, while for ACCESS-M, $r_{\mathrm{PIC}}(1-\sigma_{\mathrm{DOM}}) = 0.77\,\%$ and $w_{\mathrm{PIC}}/\tau_{\mathrm{PIC}} = 2100\,\mathrm{m}$. The rain ratio is the most sensitive, with $\Delta_{\mathrm{circ}}(r_{\mathrm{PIC}}) = 86\,\%$.

Key to the strength of the biological pump are DOM and POM export, which are controlled by a number of optimized parameters: the fraction $\sigma_{\mathrm{DOM}}$ of production allocated to DOM, the fraction $\sigma_{\mathrm{f}}$ of POP allocated to fast-sinking particles, and

the POC respiration-rate amplitudes $\gamma_{\mathrm{f}}$ and $\gamma_{\mathrm{s}}$, which are themselves dependent on temperature and oxygen via $q_{10}$, $K_{\mathrm{O_2}}$, $T_{\mathrm{ref}}$, and $[\mathrm{O_2^{lim}}]$. Each of these parameters is strongly dependent on circulation with $\Delta_{\mathrm{circ}}$ ranging roughly from 30 to 100 % (notably $\gamma_{\mathrm{f}}$ and $\gamma_{\mathrm{s}}$ have $\Delta_{\mathrm{circ}} \geq 80\,\%$). These large sensitivities despite identical observational constraints show that the biological pump operates differently in the OCIM2 and ACCESS-M circulations.





Given that parameters can be expected to be least biased when optimized for the data-assimilated OCIM2 circulation, how well does ACCESS-M PCO2 match observations when solved with OCIM2-optimized parameters? With optimized OCIM2 parameters the fidelity of the ACCESS-M tracers to observations is strongly degraded. Oxygen is most affected (RMSE doubles to $64\,\mu\mathrm{M}$) and particularly unrealistic in the Pacific where the tropical upper ocean and the old waters of the deep Pacific are strongly deoxygenated. At the tropical surface this occurs because an increased fraction of production, which is largely unaffected, is routed to DOC. (The OCIM2-optimized $\sigma$ is more than twice the ACCESS-M-optimized $\sigma$.) POM respiration, however, is shifted to greater depth because the respiration amplitudes ($\gamma_\mathrm{f}$ and $\gamma_\mathrm{s}$) are reduced relative to their ACCESS-M optimized values, which allows POM to sink deeper (greater transfer efficiency). As a result of deeper POM respiration (relative to the optimized state), oxygen is stripped out of Antarctic Bottom Water (AABW), which greatly expands the volume of hypoxic waters in the Pacific. For DIP, DIC, and TA, we find RMSEs of 0.32, 53, and $36\,\mu\mathrm{M}$, respectively, which is 30–60 % worse than the ACCESS-M optimized fit. While non-optimal parameters by definition degrade the fit to observations, these large increases in mismatch with observations underline the central role of circulation biases.

## 3.3 Biological pump

Can the optimized PCO2 model robustly predict the patterns and strength of the ocean's biological pump regardless of whether we use the OCIM2 or ACCESS-M estimates of the current ocean state? To address this question, we consider a number of simple metrics of the biological pump and contrast the OCIM2 and ACCESS-M cases.

### 3.3.1 POC flux and export production

A commonly used metric of the biological carbon pump is the POC flux through a given depth horizon. Although the POC flux through the base of the euphotic zone is arguably more robust (Buesseler and Boyd, 2009), here we simply consider the POC flux through $100\,\mathrm{m}$ depth and then consider export production (referenced to the base of the euphotic zone), which is a more robust comprehensive metric of export.

Figure 4 shows maps and global zonal integrals of the 100-m POC flux. The geographic patterns of the OCIM2 and ACCESS-M 100-m POC fluxes are broadly similar, but the globally integrated flux of $22\,\mathrm{PgC\,yr^{-1}}$ for ACCESS-M is three times larger than the $7.4\,\mathrm{PgC\,yr^{-1}}$ flux for OCIM2. Relative to OCIM2, the ACCESS-M POC flux is too large in the subtropical gyres, indicating too much production fueled by excessive DIP supply. The ACCESS-M POC flux is also larger in the subpolar oceans, particularly in the Pacific and Indian sectors of the Southern Ocean and in the North Atlantic along the Gulf Stream trajectory. These differences are likely due to the ACCESS-M model's deeper mixed layers (Fig. C1).

Because carbon can be exported in both particulate and dissolved form, a more comprehensive measure of export is the export production, that is, the rate of organic-matter production in a given euphotic water column that results in respired DIC anywhere in the aphotic ocean (Primeau et al., 2013, Appendix D). Figure 4 also shows maps and global zonal integrals of export production. Globally integrated export production, which includes export of DOM, is $16\,\mathrm{PgC\,yr^{-1}}$ for OCIM2 and $36\,\mathrm{PgC\,yr^{-1}}$ for ACCESS-M, considerably larger than the 100-m POC fluxes. The geographic pattern of export production



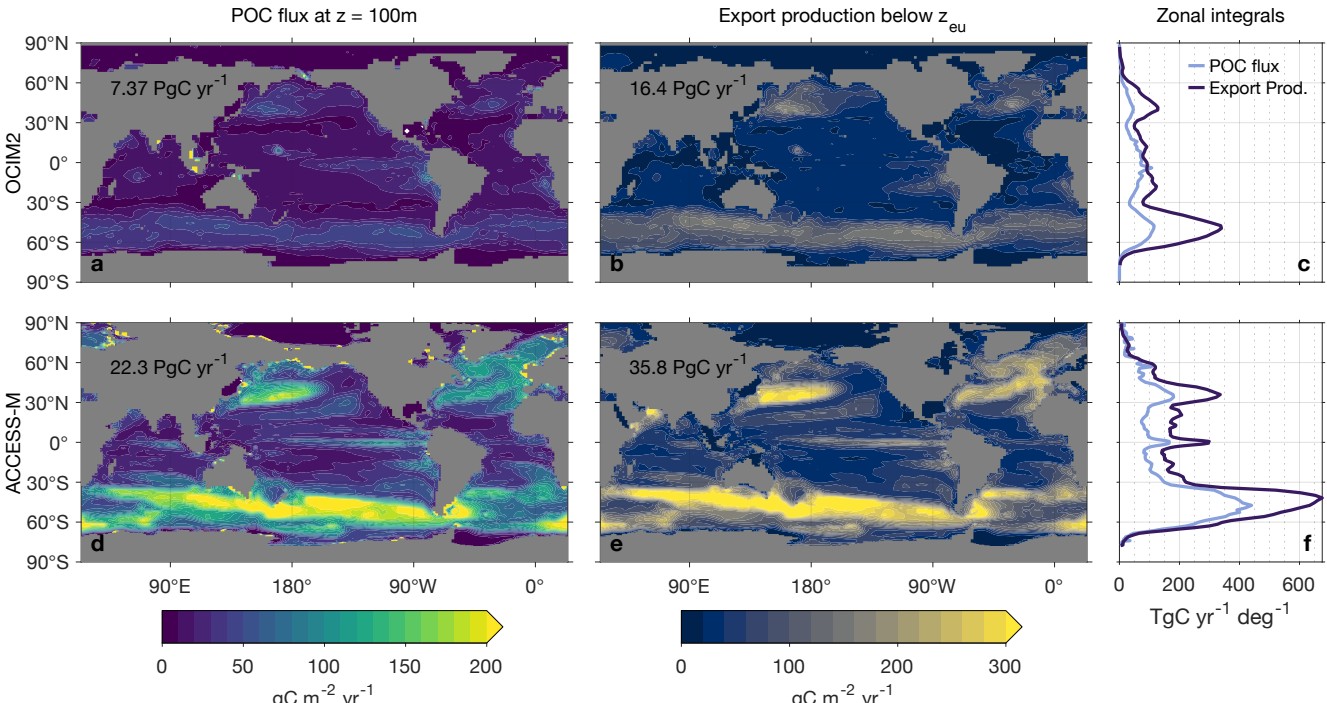

**Figure 4.** Maps and global zonal integrals of the 100-m POC flux and the carbon export production out of the euphotic zone for the PCO2 embedded in OCIM2 (top) and ACCESS-M (bottom). Global integrals are indicated on Asia.

is generally similar to that of the 100-m POC flux, but for OCIM2 subpolar export is much larger than low-latitude export in terms of export production than in terms of the 100-m POC flux.

### 3.3.2 Pump strength and regeneration pathways

A simple metric of the strength of the biological pump is the fraction of the global phosphate inventory that is regenerated,

$E_{\mathrm{P}} \equiv \langle[\mathrm{DIP}_{\mathrm{reg}}]\rangle / \langle[\mathrm{DIP}]\rangle$, where the angle brackets denote a global volume-weighted integral. We define $[\mathrm{DIP}_{\mathrm{reg}}]$ here as the concentration of DIP that was remineralized in the aphotic ocean and has not been in contact with the euphotic zone since (Appendix D2). By contrast, preformed DIP is transported out of the euphotic zone without passing through the biological pump. $E_{\mathrm{P}}$ was introduced by Ito and Follows (2005) as a metric of pump efficiency (denoted by $P^*$), but regional variations of C:P in POM complicate this interpretation prompting alternate directly carbon-based metrics of pump efficiency (e.g., Holzer

et al., 2021b). Remarkably, despite the biases of the ACCESS circulation, the OCIM2 and ACCESS-M embedded PCO2 have almost identical pump strengths of $E_{\mathrm{P}} \approx 44\,\%$ and 43 %. These values are within the 39–50 % range of previous inverse models (DeVries et al., 2012; Primeau et al., 2013; Pasquier and Holzer, 2016; Holzer et al., 2021b), but above the original AOU-based estimate of 36 % by Ito and Follows (2005).





**Figure 5.** (a) $DIP_{reg}$ contributions from $POP_f$, $POP_s$, and DOP (red, orange, blue) for the PCO2 model embedded in OCIM2 represented as both a pie chart and a bar chart. Pump strength as a percentage is indicated above the pie chart. Each bar represents the regenerated DIP inventories in terms of the corresponding export rate (width) × bulk re-exposure time (height). (b) As (a) but for ACCESS-M. (c–d) As (a–b) but for DIC, including the additional contributions from PIC (gray).

Does the biological pump operate in the same way for OCIM2 and ACCESS-M? Phosphate can be regenerated through three
pathways in our model: remineralization of $POP_f$, $POP_s$, or DOP. Similarly, DIC can be regenerated through the respiration of $POC_f$, $POC_s$, and DOC, and additionally through the dissolution of PIC (carbonate pump). To quantify the importance of each pathway, regenerated DIP and DIC are partitioned using a Green-function approach (Appendix D2). The pie charts of Fig. 5 show that the dominant contribution comes from biogenic particles, accounting for roughly 73–78 % of regenerated DIP (and hence $E_P$) and 82–84 % of regenerated DIC, for both circulations. For regenerated DIC, PIC dissolution makes a sizable




contribution of 25 % for OCIM2 and 23 % for ACCESS-M, consistent with the very deep dissolution of PIC (exponential
profile with $e$-folding length of 3490 m for OCIM2 and 2060 m for ACCESS-M).

To better understand how the biological pump sets the size of the regenerated DIP and DIC pools, it is useful to think about
the bulk sequestration time of the regenerated pool and the corresponding export rates. We therefore write the regenerated
inventory (for a given mechanism) as the product of the corresponding globally integrated export production and the corre-
sponding bulk sequestration time. (Bulk sequestration/residence time is simply defined here as the ratio of inventory over rate
and is thus equal to the regeneration-weighted mean water-reexposure time.) The bulk sequestration times and corresponding
export productions for each regeneration mechanism are plotted as boxes in Fig. 5, with the height of the box being the se-
questration time, the length being the export production, and the area being proportional to the regenerated inventory. Despite
similar POM-regenerated pools, the export production rates from POM are roughly 3× larger for ACCESS-M than for OCIM2,
compensating for 3× shorter sequestration times. (This is the case for all POM types, whether it be POP or POC, slow or fast.)
For ACCESS-M, strong export rates (wide boxes) are due to rapid uptake (large $p_{max}$) and deep mixed layers, while short
sequestration times (short boxes) are due to rapid (large $\gamma_s$ or $\gamma_f$), and thus shallow, respiration. This is a striking example
of how parameter optimization can change the inner workings of the biological pump to compensate for transport biases. We
hasten to add, however, that we do not use POM measurements as a constraint on the model so that the relative contributions
due to slow and fast POM are likely model specific.

The smaller optimized PIC:POC ratio for ACCESS-M ($r_{PIC} = 1.02$ % compared to 6.28 % for OCIM2; Table 1) com-
pensates ACCESS-M's larger carbon export, resulting in ACCESS-M and OCIM2 having similar PIC exports (0.82 and
0.73 PgC yr$^{-1}$). We note that while the value of $r_{PIC} = 1.02$ % is optimal for ACCESS-M, it is unrealistically small com-
pared to other estimates that range from roughly 3 to 12 % (Sarmiento et al., 2002; Jin et al., 2006; Kwon et al., 2022). The
sequestration times of the PIC-regenerated DIC pools are also similar for the two circulation cases (670 yr for OCIM2, 540 yr
for ACCESS-M) despite widely different PIC sinking speeds (and thus dissolution depths given the fixed dissolution timescale
$\tau_{PIC}$). This points to compensation due to subtle differences in the regeneration-weighted water re-exposure times. Overall, the
carbonate pump contributes about a quarter of the global regenerated DIC inventory, regardless of circulation. The robustness
of PIC export and PIC-regenerated DIC sequestration times and inventories across circulations is likely due to the alkalinity
constraint, which tends to adjust the PIC pump to match TA observations.

Figure 5 shows that DOM remineralization makes a substantial contribution to the regenerated DIP and DIC inventories.
The sequestration times of DOM-regenerated DIC and DIP are only a few decades as expected given the short 2-yr $e$-folding
time for semilabile DOM in our model. The sequestration time of DOM-regenerated DIP or DIC for OCIM2 is ∼1.8 times
larger than for ACCESS-M, but its export rate is ∼1.5 times smaller, giving roughly comparable DOM-regenerated DIC pools
for both circulations. The larger DOC export for ACCESS-M is consistent with its larger nutrient and carbon uptake, in turn
consistent with its deeper mixed layer supplying more nutrients. We emphasize that DOP and DOC were modelled very simply
here with a single uniform lifetime and that we did not use any DOM observational constraints (which would require multiple
DOM tracers with a spectrum of labilities). Thus, while our diagnostics demonstrate that DOM can be an important contributor
to export production, the specific values of the DOM-driven export obtained here cannot be considered to be accurate for the



real ocean. With OCIM2, DOM accounts for roughly 50 % of the export production while recent work places the contribution of DOM to carbon export at around 20 % (Letscher et al., 2015).

### 3.3.3  POC transfer efficiency

The different ways in which OCIM2 and ACCESS-M PCO2 achieve optimum fits to the observations are also manifest in the models' particle dynamics, examined here in terms of the POC transfer efficiency. The efficiency of POC transfer from depth
$z_1$ to a deeper depth $z_2$ is simply the ratio $\Phi(z_2)/\Phi(z_1)$ where $\Phi(z)$ is the POC flux at depth $z$. The transfer efficiency is a convenient and observable metric of POC flux attenuation with depth: High efficiency corresponds to lower respiration rates and hence to particles surviving to greater depth.

Figure 6 shows maps of the POC transfer efficiency from the base of the euphotic zone ($z_1 = z_{\mathrm{eu}}$) to 500 m deeper ($z_2 = z_{\mathrm{eu}}+500\,\mathrm{m}$) together with [O$_2$] averaged over the $z_1$-to-$z_2$ water column. For OCIM2, the transfer efficiencies of both slow and
fast particles have patterns that have a strong inverse correlation with the low oxygen concentrations of the Pacific OMZs. At a given temperature, respiration rates are modulated by the [O$_2$] Michaelis-Menten factor in Eq. (2), so that lower oxygen and respiration rates result in higher transfer efficiency, as expected. The fast-sinking POC$_{\mathrm{f}}$ achieves a 500-m transfer efficiency of 0.75 in the global mean with local values over 0.85 in the Pacific OMZs. The slow-sinking POC$_{\mathrm{s}}$ has more time to respire over a given depth range and hence has a transfer efficiency of only 0.08 in global mean, reaching around 0.25 in the Pacific
OMZs. The transfer efficiencies are elevated by around 0.05 in high latitudes because of lower respiration in colder waters as parameterized by the $q_{10}$ term in Eq. (2).

We note in passing that the reduced respiration in cold waters competes with increased seawater viscosity, which slows sinking particles down (smaller viscosity factor; see Appendix A3). The slower sinking allows for respiration to act over a longer period of time, compensating for the lower respiration rates. Depending on the value of $q_{10}$, this compensation could
in principle erase the temperature dependence of respiration. However, for both models, parameters optimize such that the compensation is only partial, with the effect of reduced respiration dominating the effect of increased viscosity. The compensation is stronger for OCIM2 PCO2, for which the viscosity effect is empirically equivalent to dividing temperature (in °C) by roughly a factor of 2.4 in the $q_{10}$ term, compared to a corresponding factor of only about 1.3 for ACCESS-M PCO2.

The spatial patterns of the transfer efficiency for both fast and slow POC are markedly different for OCIM2 and ACCESS-
M. The different patterns are a consequence of the different optimal respiration parameters $K_{\mathrm{O}_2}$ and $q_{10}$. For both slow and fast POC, the highest transfer efficiencies for ACCESS-M occur in subpolar and polar waters, because of the much greater sensitivity to temperature (about twice as large a value of $q_{10}$ compounded with weaker viscosity compensation). In terms of contributions to the regenerated DIC inventory, we note that the deeper POC respiration in the ACCESS-M Southern Ocean is compensated in part by the shorter re-exposure times of about 200 yr (Holzer et al., 2020) compared to up to 700 yr for OCIM2
(DeVries and Holzer, 2019). For ACCESS-M, the temperature dependence dominates the oxygen dependence with $K_{\mathrm{O}_2}$ being 2.5 times smaller than for OCIM2. Compared to the OCIM2 PCO2 model, oxygen in ACCESS-M must therefore drop to 2.5 times lower concentrations for the same reduction in respiration, which is a bit more likely to occur because of ACCESS-M's





**Figure 6.** The efficiency of transferring slow-sinking POC (top) and fast-sinking POC (middle) from the base of the euphotic zone at depth $z_{eu}$ to depth $z_{eu} + 500\,\text{m}$ together with the oxygen concentration averaged over the transferred depth range (bottom) for OCIM2 (left) and ACCESS-M (right).

lower OMZ oxygen concentrations (Figures 1 and 2). As a result, ACCESS-M transfer efficiencies still show enhancement in the OMZs by about 0.3 for fast POC and only 0.03 for slow POC.

### 3.4 Response to Southern Ocean nutrient drawdown

Given optimal parameters for both embedding circulations, how robust is PCO2's response to perturbations? Motivated by previous studies that explored the importance of the iron-limited Southern Ocean as a source of preformed nutrients to the rest of the ocean (e.g., Sarmiento et al., 2004; Marinov et al., 2006; Primeau et al., 2013; Holzer and Primeau, 2013; Holzer et al., 2021b), we perturb the system by forcing nearly complete nutrient utilization south of 30 °S. This is accomplished by adding a



DIP uptake rate of the form $[\text{DIP}]/\tau^*$ with $\tau^* = 0.1\,\text{d}$. In the following we contrast the ensuing responses of the OCIM2 and

ACCESS-M embedded nutrient, carbon, and oxygen cycles.

     Our idealized perturbation increases carbon uptake south of $30\,°\text{S}$ by 260 % for OCIM2 and by 360 % for ACCESS-M. This

achieves nearly complete nutrient utilization south of $30\,°\text{S}$, which redistributes DIP (phosphate) globally because the total

amount of phosphate is conserved in our formulation. Phosphate becomes "trapped" in the Southern Ocean (Primeau et al.,

2013), reducing nutrient concentrations north of $30\,°\text{S}$, where biological production is reduced by 25 % for OCIM2 and by

30 % for ACCESS-M. The dramatic production increase in the Southern Ocean cranks up the global biological pump strength

$E_\text{P}$ to almost 90 % for both circulations, similar to findings of Primeau et al. (2013).

     To visualize the global redistribution of nutrients, Fig. 7 shows the basin zonal averages of the DIP response. For OCIM2

PCO2, intense Southern Ocean nutrient trapping is evident with [DIP] increases of up to $1\,\mu\text{M}$ at depth. The depletion of

surface nutrients south of $30\,°\text{S}$ deprives Antarctic Intermediate and Mode Waters (AAIW and AAMW) of the preformed DIP

that they supply in the unperturbed state to the rest of the ocean (e.g., Sarmiento et al., 2004). The abyssal branch of the

overturning circulation (Holzer et al., 2021a) extends elevated Southern Ocean DIP concentrations into the abyssal Pacific and

Indian oceans. In the Atlantic, the nutrient trapping is more confined to high southern latitudes with North Atlantic Deep Water

(NADW) still supplying up to $0.5\,\mu\text{M}$ preformed DIP in the zonal mean (not shown).

For ACCESS-M, the Southern Ocean nutrient trapping is less intense than for OCIM2. The contrast with OCIM2 is partic-

ularly striking in the Atlantic sector where increases in DIP barely reach a quarter of the OCIM2 DIP response. The reason for

this contrast lies in ACCESS-M's unrealistically deep mixed layers in the Weddell and Ross Seas (Appendix C), which are not

present for OCIM2. The nutrient trapping in OCIM2 occurs by POP being regenerated deep in upwelling Circumpolar Deep

Water (CDW), which returns DIP with the surface Ekman divergence to regions of high production where the POP flux to

depth is maintained. In ACCESS-M, this sinking-POP–upwelling-DIP trapping loop is short circuited in the regions of unreal-

istic deep mixing. In these regions, DIP regenerated at depth is quickly mixed throughout the water column and utilized again

instead of being slowly returned to the surface by upwelling CDW. Because DOP is not bioavailable in our model, phosphorus

in the high-mixing regions is in effect siphoned out of the deep-mixing regions as DOP, with little remaining as DIP. (Outside

the deep mixing regions, regenerated DIP is trapped by the same mechanism as for OCIM2.) The zonal-mean DIP depletion

due to the short-circuit siphon is visible in Figs. 8g and 8k south of $60\,°\text{S}$ in the Atlantic and Pacific, with weaker depletion

in the Pacific where deep mixing occupies a smaller fraction of the basin. The weaker Southern Ocean nutrient trapping for

ACCESS-M results in a correspondingly weaker response north of $60\,°\text{S}$: Above $1000\,\text{m}$, ACCESS-M has smaller decreases in

preformed DIP (not shown) carried northwards by surface currents, AAMW, and AAIW, and at depth ACCESS-M has smaller

increases in regenerated DIP (not shown) extending northwards with the abyssal overturning of the Pacific and Indian Oceans.

The DIC response shown in the basin zonal means of Fig. 8 is the result of both Southern Ocean nutrient trapping and

changes in air–sea $CO_2$ exchange. Similar to the DIP response, DIC trapped in the Southern Ocean propagates northwards at

depth with AABW for both OCIM2 and ACCESS-M. The DIC response is generally larger than would be expected from the

DIP response using the C:P stoichiometry of POM (which for zero DIP saturates at about $167\,\text{molC}\,\text{molP}^{-1}$, Galbraith and

Martiny, 2015). The extra DIC is supplied by $CO_2$ ingassing driven by the strong surface DIC drawdown, which changes the





**Figure 7.** Atlantic, Pacific, and Indian Ocean zonal-mean DIP responses to complete nutrient drawdown south of 30 °S. (a–c) Perturbed DIP for OCIM2 PCO2. (d–f) Corresponding difference between perturbed DIP and base (unperturbed) DIP. (g–l) As (a–f) but for ACCESS-M PCO2.

Southern Ocean from net $CO_2$ outgassing to net $CO_2$ ingassing. The global DIC inventory increases by roughly 7 % for both circulations. For both OCIM2 and ACCESS-M, the Southern Ocean $CO_2$ ingassing weakens the decrease of preformed DIC (due to intensified uptake) that is propagated via AAMW and AAIW such that the total DIC response is dominated by the response of regenerated DIC. As for DIP, Southern Ocean DIC trapping is more pronounced for OCIM2 than for ACCESS-M, which is again a consequence of ACCESS-M's unrealistic deep mixing in the Weddell and Ross Seas.

The zonal mean $[O_2]$ response quantified in Figure 9 shows less sensitivity to the choice of circulation. For both OCIM2 and ACCESS-M, intensified Southern Ocean production dramatically deoxygenates the ocean driving $[O_2]$ in Southern Ocean sourced water masses (SAMW, AAIW, AABW) to near zero. (The prominent oxygen plume that can be seen in the deep





**Figure 8.** Atlantic, Pacific, and Indian Ocean zonal-mean DIC responses to complete nutrient drawdown south of 30 °S. (a–c) Perturbed DIC for OCIM2 PCO2. (d–f) Corresponding difference between perturbed DIC and base (unperturbed) DIC. (g–l) As (a–f) but for ACCESS-M PCO2.

South Indian Ocean (Fig. 9c and i) is fed by CDW propagating eastward from the Atlantic.) This deoxygenation is driven by strongly increased respiration which balances the strongly increased Southern Ocean organic-matter production. Strongly

increased dissolved organic matter propagates northward from the Southern Ocean with SAMW, AAIW, and AABW, shaping the oxygen response seen in Fig. 9. Increased photosynthetic oxygen production in the Southern Ocean increases $[O_2]$ near the surface, most of which is quickly lost to the atmosphere and thus not able to meet the greater oxygen demand at depth. Outside of the Southern Ocean, oxygen weakly increases near the surface (by up to 50 μM) and in northern NADW consistent with decreased production north of 30 °S and decreased respiration in NADW, which also manifested in decreased DIC (Fig. 8d,j).





**Figure 9.** Atlantic, Pacific, and Indian Ocean zonal-mean $O_2$ responses to complete nutrient drawdown south of 30 °S. (a–c) Perturbed $O_2$ for OCIM2 PCO2. (d–f) Corresponding difference between perturbed $O_2$ and base (unperturbed) $O_2$. (g–l) As (a–f) but for ACCESS-M PCO2.

Because the overall oxygen response is dominated by increased respiration driving $[O_2]$ to near zero in much of the ocean, the difference in the trapping mechanisms between OCIM2 and ACCESS-M is not as manifest in Fig. 9. (For both circulations, most of the deep southern, Indian, and Pacific oceans become OMZs as the global oxygen content is reduced by about 60 %.) Nevertheless, for ACCESS-M stronger oxygen decreases in the deep Southern Ocean and weaker vertical gradients south of 60 °S still show the effect of the rapid deep mixing in ACCESS-M. The more rapid vertical exchange with the surface oxygen

supply in the Ross and Weddell Seas for ACCESS-M prevent the Atlantic and Pacific south of 60 °S from being as strongly deoxygenated as in OCIM2.



It is worth noting that the response to Southern Ocean nutrient drawdown is completely dominated by the circulation. Indeed, solving ACCESS-M PCO2 with OCIM2 optimal parameters results in responses that are nearly the same as those shown in Figures 7–9.

## 455   4   Discussion

This study was motivated by the challenges posed in using ocean circulations from climate models to capture the workings of the biological pump and its effect on the ocean's oxygen distribution. In particular, how do biases in a circulation model for the current state of the ocean affect our ability to match observations, and if model parameters are optimized to match observations as well as possible, how do circulation biases affect the response of the biological pump to perturbations? Answers to these

questions are important for assessing predictions for the future biogeochemical state of the ocean.

To address these issues, we built a model (PCO2) of the coupled nutrient, carbon, and oxygen cycles. The mechanistic uptake of PCO2 and the simpler treatment of DOC that this affords are the key differences between PCO2 and the SIMPLE-TRIM model of DeVries and Weber (2017), which otherwise share essentially the same formulation of POM respiration. The fully interactive oxygen of PCO2 is the key difference with OCMIP-style models (Najjar et al., 2007) for which POM flux-

divergence profiles are prescribed and organic carbon passes through the semilabile DOC pool before being respired (e.g., Holzer, 2022).

We modelled a minimal set of biogeochemical tracers (PO4, POP, DOP, DIC, POC, DOP, PIC, O2, TA), in part because of the greater computational demands of the ACCESS-M circulation (even when coarse-grained to nominal $2° \times 2°$ horizontal resolution). In particular, we use only a single semilabile pool of DOC, as opposed to separate labile, semilabile, and recalcitrant

pools (e.g., DeVries and Weber, 2017; Kwon et al., 2022). For this single DOC pool, remineralization is modelled using a simple fixed 2-yr $e$-folding because on one hand we lack quantitative estimates of its biological and photochemical degradation and on the other hand neglect of labile and refractory DOC pools justifies a simpler representation of semilabile DOC. By the same token, for simplicity neither DOC nor DOP are bioavailable in our model, although in the real ocean DOP provides phosphorus to P-limited phytoplankton in highly oligotrophic regions (e.g., Letscher and Moore, 2015).

The absence of an explicit nitrogen cycle means that we cannot make detailed statements about how denitrification might be affected by circulation changes, but the basic effect of organic-matter oxidization continuing in anoxic regions is parameterized. We do not model dissolved iron either, but PCO2 captures the large-scale patterns of production because uptake parameters are optimized against observed nutrient distributions, which are shaped by all processes in the real ocean. (E.g., there is more phosphate in the Southern Ocean than there would be without iron limitation.) We justify these approximations *a posteriori* by

the high quality of the fit to the observations for the data-assimilated OCIM2 circulation.

For most parameters the relative variation of the optimal value with circulation is larger than the variation with model complexity (meaning prescribed versus simulated oxygen here), underlining the all-important control of transport on ocean biogeochemistry. Our findings also show that caution is necessary when comparing parameter values among models. Unless the circulation is free from biases and the formulation of a given process can be justified from fundamental biology and



chemistry, parameter optimization is not the same as the estimation of fundamental parameters, the value of which could in principle be measured directly. Instead, optimized model parameters take on values that tend to compensate for shortcomings of the circulation and biogeochemical formulation.

Even when optimized with the data-assimilated OCIM2 circulation, significant biases in the biogeochemical tracers remain, pointing to model deficiencies. Remarkably, for the OCIM2 circulation, the remaining biases in the oxygen distribution are similar to those of a much simpler OCMIP-style model of oxygen also embedded in OCIM2 (Holzer, 2022). This points to potential issues with the OCIM2 circulation, air–sea exchange, and/or the parameterization of the oxygen dependence of microbial respiration. An important caveat that must be kept in mind is that the covariance between biological production, air–sea exchange, and seasonally varying circulation is not captured with our steady circulation models. In particular, the models specify the mixed-layer depth to be the climatological annual maximum, which could over-oxygenate high-latitude regions consistent with the OCIM2 optimized state having too much oxygen in the mid-depth subpolar North Pacific.

Remaining model biases could potentially be reduced by using additional observational constraints. For example, POC observations (e.g., Dinauer et al., 2022) could be used to better constrain particle dynamics, although these are currently only available for a very sparse set of stations. Also, our results show that DOC transport is an important pathway for carbon export, suggesting potential value from using DOC observations as constraints. However, typically, total DOC is measured, not just the semi-labile fraction, so that modelling all DOC pools becomes necessary, which would increase computational cost considerably. One could also try to constrain nutrient uptake with satellite phytoplankton measurements, but this would entail using different functional classes (e.g., Pasquier and Holzer, 2017) and hence again lead to greater model complexity, and the larger set of parameters would make the optimization more costly.

The matrix formulation of PCO2 not only allowed for efficient steady-state solutions, and hence parameter optimization, but also enabled us to diagnose the inner workings of the biological pump. For example, partitioning regenerated DIP according to which mechanisms produced it is generally not computationally feasible for forward models, for which regenerated DIP is typically approximately inferred from apparent oxygen utilization (Ito and Follows, 2005, e.g.,) or estimated by computing preformed DIP and calculating $DIP_{reg}$ as a residual (Marinov et al., 2008, e.g.,). Here we were able to calculate this partitioning in steady state directly and accurately as detailed in Appendix D2. Similarly, export production is computationally prohibitive for forward models, but readily available in the matrix formulation using an adjoint approach (Appendix D1).

## 5 Conclusions

To explore the effects of climate-model circulation biases on the global biological pump, we embedded a steady-sate model of the ocean's nutrient, carbon, and oxygen cycles (PCO2) in the ACCESS-model-derived decadal-mean ocean circulation for the 1990s and contrasted the results with PCO2 embedded in the data-assimilated OCIM2 circulation model. The differences between the OCIM2 and ACCESS cases in optimized biogeochemical parameters and in their responses to Southern-Ocean nutrient drawdown lead us to the following main conclusions:



With optimized parameters, the PCO2 model is able to match the observed DIP, DIC, $O_2$, and TA fields with reasonable fidelity for both circulations, despite some strong biases in the ACCESS circulation. However, the fit for the ACCESS circulation is not as good as for OCIM2, with RMSEs that are roughly 1.5–2.5 times larger. Neither circulation captures all the features of the $O_2$ distribution. In OMZs, the oxygen concentration is overestimated for OCIM2 and underestimated for AC-CESS, which points to biases in both models (possibly in both biogeochemistry and circulation) that are not compensated by parameter optimization. Simulating $[O_2]$ as compared to prescribing it from observations, increases the RMSEs for the other tracers regardless of the circulation model.

The parameter values optimized for the realistic data-assimilated OCIM2 circulation are not optimal for the ACCESS-M-embedded biogeochemistry. Optimal parameter values vary by up to a factor of 7 between OCIM2 and ACCESS-M and using OCIM2 parameters for ACCESS-M degrades the fit by 30–60 %. This is in agreement with the findings of Kriest et al. (2020), that "one size does not fit all". Circulation is a key control on biogeochemistry, with optimal parameter values varying more with circulation than with the complexity of the biogeochemistry model (Matear and Holloway, 1995).

Despite fitting observed tracers reasonably well, the optimized biological pump operates differently in the two circulations. This manifests in the ACCESS-M export production being roughly two times larger and its 100-m POC flux being roughly three times larger than for OCMI2, which has a $7.4\,\mathrm{PgC\,yr^{-1}}$ 100-m POC flux and a $16.4\,\mathrm{PgC\,yr^{-1}}$ export production. Despite these large export differences, the biological pump strength (quantified by the regenerated fraction of the DIP inventory) is robust at 43–44 % across embedding circulations. Between a third and a half of the global production is exported as DOC, which contributes less than a fifth of the regenerated DIC inventory for both OCIM2 and ACCESS-M. The remaining carbon export is provided mainly by POC, with PIC contributing just a few percent.

Widely different exports with similar pump strengths are reconciled by differences in sequestration times (DeVries et al., 2012; Holzer et al., 2021b). We find that DOC- and POC-regenerated DIC takes roughly three times as long to return to the euphotic zone for OCIM2 than for ACCESS-M so that OCIM2 has a higher sequestration efficiency: a smaller export rate acts over a longer time resulting in similarly sized pools of respired carbon. For the carbonate pump (PIC), deep dissolution leads to a sequestration time of roughly $600\,\mathrm{yr}$ and accounts for almost a quarter of the regenerated DIC inventory for both circulations.

Differences in particle dynamics shape differences in the biological pump. Globally, POC is respired deeper in OCIM2 compared to ACCESS-M, but regionally the largest differences in transfer efficiency occur in OMZs and at high latitudes through the oxygen and temperature dependence of respiration. For OCIM2, respiration is optimized to have a weak temperature but a strong oxygen dependence, enhancing transfer efficiency primarily in OMZs. For ACCESS-M, temperature dominates variations in respiration enhancing transfer efficiency mostly in high latitudes. In the ACCESS-M Southern Ocean, deeper remineralization is counteracted by much shorter deep re-exposure times (less than $200\,\mathrm{yr}$ compared to up to $700\,\mathrm{yr}$ for OCIM2) resulting in similar global pump strengths.

Despite PCO2 fitting observed tracers, the differences in the biological pump drive differences in the response to Southern Ocean nutrient drawdown. For OCIM2, the strongly stimulated Southern Ocean production leads to intense nutrient trapping and increased carbon sequestration in the deep oceans. For ACCESS-M, the Southern Ocean nutrient trapping is partially short circuited by rapid vertical mixing in the unrealistically deep mixed layers of the Weddell and Ross Seas, where the intensified



surface production siphons DIP out of the entire water column. The DIC response is broadly similar to the DIP response, but outside of the Southern Ocean DIC is not as depleted and there is enhanced DIC leakage with mode/intermediate waters due to enhanced C:P and $CO_2$ ingassing in the Southern Ocean. Southern Ocean nutrient drawdown leads to nearly complete

deoxygenation of Southern Ocean sourced water masses: For both circulations, strongly increased POC and DOC production leads to sharply increased oxygen demand that cannot be met by increased ocean photosynthesis. Because $[O_2]$ is almost driven to zero over much of the deep ocean, differences in the $[O_2]$ responses between the two circulations are only pronounced south of $\sim 60\,°S$ where the rapid deep mixing in ACCESS-M provides better oxygenation.

Our findings show that optimizing biogeochemical parameters to match observed tracers does not guarantee a robust rep-

resentation of the biological pump. Biases in the circulation influence how the biological pump operates and its response to perturbations, even when parameters are optimized to match biogeochemical tracer fields. It is thus imperative to quantify the inner workings of the biological pump in biogeochemical models to assess the response of the carbon and oxygen cycles to climate change.

*Code and data availability.*   The MATLAB code for this work can be found at [Zenodo archive URL; to be updated at publication]. The

ACCESS-M transport matrix was built from the "historical" ACCESS-1.3 CMIP5 model runs, which are available at https://esgf.nci.org.au/ projects/esgf-nci/ and also include temperature, salinity, sea-ice, and wind fields. The irradiance (photosynthetically available radiation) fields for both the ACCESS-M- and OCIM2-embedded PCO2 are also from the ACCESS1.3 runs. The OCIM2 transport matrix and corresponding salinity and temperature fields are available at https://tdevries.eri.ucsb.edu/models-and-data-products/. For OCIM2, sea-ice and surface-wind data are from National Centers for Environmental Prediction (NCEP) re-analysis available at https://psl.noaa.gov/data/gridded/data.

ncep.reanalysis.html. The gridded phosphate and silicate observations are from the World Ocean Atlas 2018 (Garcia et al., 2019) available at https://www.ncei.noaa.gov/access/world-ocean-atlas-2018. The gridded DIC, $O_2$, and TA observations are from GLODAPv2 (Lauvset et al., 2016) available at https://www.glodap.info/index.php/mapped-data-product/.

## Appendix A: Biogeochemistry Model Details

### A1   Biological Phosphate Uptake

Following Pasquier and Holzer (2017), phosphate uptake $U_P$ is parameterized as a function of temperature, light, and nutrient availability:

$$U_\mathrm{P} \equiv \frac{p_\mathrm{max}}{\tau_U}\, e^{\kappa_T T} \left( \frac{I}{I + K_I} \right)^2 \left( \frac{[\mathrm{DIP}]}{[\mathrm{DIP}] + K_\mathrm{DIP}} \right)^2. \tag{A1}$$

In Eq. (A1) $T$ is the Celsius temperature, and $I$ is the photosynthetically available radiation field. $I$ is taken from the ACCESS1.3 model runs and was applied to both the ACCESS-M- and OCIM2-embedded PCO2 models. The main difference

with the work of Pasquier and Holzer (2017) is that explicit iron and silicate limitation are not included for simplicity. Phosphate uptake is modeled to have exponential temperature dependence following Eppley (1972) who tuned $\kappa_T = 0.063\,\mathrm{K}^{-1}$,



which was later validated statistically (e.g., Bissinger et al., 2008). Light availability and nutrient limitation are parameterized as Monod factors, the square of which controls the strength of phosphate utilization following the logistic phytoplankton growth model used by Pasquier and Holzer (2017) and broadly inspired by Galbraith et al. (2010). The optimized parameter $p_{\max}$ represents the phytoplankton concentration for nutrient and light replete condition (unit Monod factors) and $T = 0\,°C$, while $\tau_U$ is a nominal uptake timescale set to 30 days.

## A2 Uptake C:P Stoichiometry

The C:P stoichiometry of biological production has been shown to have strong regional deviations from the 106:1 Redfield value (e.g., Teng et al., 2014). Here we model the C:P of biological production to be identical to the C:P ratio of POM in the surface ocean, which is known to be strongly correlated with surface [DIP]. Galbraith and Martiny (2015) showed that the P:C of surface POM can be fit by the linear [DIP] dependence

$$r_{\mathrm{P:C}} = m\,[\mathrm{DIP}] + b, \tag{A2}$$

with slope $m = 6.9\,\mathrm{mmol\,mol^{-1}\,\mu M^{-1}}$ and intercept $b = 6.0\,\mathrm{mmol\,mol^{-1}}$.

By constraining the parameters $m$ and $b$ of Eq. (A2) using an OCIM2-embedded inverse model of the carbon cycle, including semi-labile and recalcitrant DOP and DOC pools and a detailed representation of PIC and riverine carbon inputs, Kwon et al. (2022) recently provided an independent verification that a linear P:C dependence on [DIP] provides a good fit to observed tracers for values of $m$ and $b$ that are consistent with the fits of Galbraith and Martiny (2015). Phytoplankton frugality in very nutrient-poor regions has been hypothesized to be modelled better by a power-law dependence of P:C on [DIP] (Tanioka and Matsumoto, 2017; Matsumoto et al., 2020), but Kwon et al. (2022) show that their inverse model is able to fit observations equally well regardless of whether a linear or power-law relationship is used. We therefore use the simpler linear relationship Eq. (A2) with $r_{\mathrm{C:P}} = 1/r_{\mathrm{P:C}}$ and the values of $m$ and $b$ from Galbraith and Martiny (2015).

## A3 Viscosity Effect on Sinking Speeds

Here, we follow a similar approach to that of Taucher et al. (2014) and define a viscosity factor $\alpha_\mu$ that multiplies the constant sinking-speed parameter ($w_{\mathrm{f}}^*$, $w_{\mathrm{s}}^*$, or $w_{\mathrm{PIC}}^*$) to obtain the local sinking speed (of POM$_{\mathrm{f}}$, POM$_{\mathrm{s}}$, or PIC). The factor $\alpha_\mu$ is given in terms of temperature ($T$) and salinity ($S$) by

$$\alpha_\mu(S,T) = \frac{\mu(S,0\,°C)}{\mu(S,T)}\,\frac{\rho_{\mathrm{p}} - \rho_{\mathrm{sw}}(S,T)}{\rho_{\mathrm{p}} - \rho_{\mathrm{sw}}(S,0\,°C)}, \tag{A3}$$

where the first term represents the effect of dynamic viscosity $\mu$ and the second term the effect of changing buoyancy ($\rho_{\mathrm{p}}$ is the particle density and $\rho_{\mathrm{sw}}$ is seawater density). For $\rho_{\mathrm{p}}$ we follow Taucher et al. (2014) and set it to a constant value of $\rho_{\mathrm{p}} = 1060\,\mathrm{kg\,m^{-3}}$, representing an average across the literature (Logan and Hunt, 1987; Bach et al., 2012). For $\rho_{\mathrm{sw}}(S,T)$ we use the MATLAB Gibbs-SeaWater (GSW) Oceanographic Toolbox (IOC et al., 2010). For dynamic seawater viscosity, $\mu(S,T)$, we use the equation of Sharqawy et al. (2010).





## Appendix B: Computational Methods

### B1 Steady-State Solver

Equations (1), (3), (4), (5) are discretized and collected into a nonlinear system of equations, $\boldsymbol{F}(\boldsymbol{x}) = 0$ where $\boldsymbol{x}$ is the con-
615 catenated vector of all the tracers. This nonlinear systems is efficiently solved using Newton or quasi-Newton methods for root
finding, which iteratively update the state vector via $\boldsymbol{x}_{i+1} = \boldsymbol{x}_i - \mathbf{J}_i^{-1}\boldsymbol{F}_i$ where $\boldsymbol{F}_i = \boldsymbol{F}(\boldsymbol{x}_i)$ and usually $\mathbf{J}_i = \mathbf{J}(\boldsymbol{x}_i)$ is the
Jacobian of $\boldsymbol{F}$ at $\boldsymbol{x}_i$. In practice, the Jacobian is factored and not updated at every iteration to save computational resources
(Kelley, 2003). Additionally, to reduce the memory required for factorization, we divide the system into smaller subsystems
for P, C, and $O_2$, and then solve the subsystems iteratively until the entire system has converged. Specifically, we first solve the
620 P subsystem (DIP, $POP_f$, $POP_s$, DOP), then the $O_2$ equation, then the C subsystem (DIC, $POC_f$, $POC_s$, DOC, PIC, TA), and
repeat until $\boldsymbol{F}(\boldsymbol{x}_i) \approx 0$.

### B2 Positivity

In practice, many equations of the PCO2 model are modified to ensure that some variables remain positive. This is useful, for
example, in Monod factors such as that of $O_2$ in Eq. (2), to avoid catastrophic cancellation (e.g., if $[O_2] \approx -K_{O_2}$ numerically).
Hence, where positivity of a variable $X$ is required, we replace $X$ with the differentiable approximation to $\max(X, 0)$ given
by

$$\max(X, 0) \approx X_0 \log(1 + e^{X/X_0}), \tag{B1}$$

where $X_0$ is carefully chosen for every variable $X$ to balance smoothness against accuracy (larger $X_0$ the a smoother but worse
the approximation). Specific values used are $X_0 = 0.1\,\mu M$ for DIP, $10\,\mu M$ for DIC and TA, and $1\,\mu M$ for $O_2$.

### B3 Smoothness

Because we are using Newton-type solvers to find the steady state of the tracer equations (B1), we replace discontinuous
or non-differentiable equations with smooth and differentiable approximations. For example, for oxygen utilization (Eq. (5)),
which is abruptly turned off for $[O_2] < 5.125\,\mu M$, we approximate the Heaviside function by

$$\Theta(X) = \frac{1}{2}\big(1 + \tanh(X/X_0)\big), \tag{B2}$$

where $X_0$ controls the smoothness and the same values of $X_0$ are used as in B2.

### B4 Objective Function

Our goal is to minimize the mismatch of the steady-state solution to Equations (1), (3), (4), (5) with the corresponding observa-
tions (DIP, DIC, TA, and $O_2$). We measure the difference of tracer $X$ with its observed values $X^{\mathrm{obs}}$ using the volume-weighted




**Table B1.** Permissible ranges for optimized parameters.

| Parameter | Range | Unit |
|---|---|---|
| $p_{\mathrm{max}}$ | 1–50 | µM |
| $K_{\mathrm{DIP}}$ | 0.01–4 | µM |
| $r_{\mathrm{PIC}}$ | 1–9 | % |
| $w^*_{\mathrm{PIC}}$ | 1800–4500 | $\mathrm{m\,d^{-1}}$ |
| $\sigma$ | 1–99 | % |
| $\sigma_{\mathrm{f}}$ | 1–99 | % |
| $\gamma_{\mathrm{f}}$ | 0.01–1.6 | $\mathrm{d^{-1}}$ |
| $\gamma_{\mathrm{s}}$ | 0.01–0.8 | $\mathrm{d^{-1}}$ |
| $q_{10}$ | 1–5 | - |
| $K_{\mathrm{O_2}}$ | 2–30 | µM |
| $r_{\mathrm{O_2:C}}$ | 1.3–1.5 | $\mathrm{molO_2\,molC^{-1}}$ |

quadratic mismatch metric

$$f_X = \frac{\int \mathrm{d}V\,(X - X^{\mathrm{obs}})^2}{\int \mathrm{d}V\,(X^{\mathrm{obs}} - \overline{X}^{\mathrm{obs}})^2}, \tag{B3}$$

where the denominator is the spatial variance of $X^{\mathrm{obs}}$, which provides a convenient scale for normalizing the misfit.

The objective function $\hat{f}(\boldsymbol{p})$ to be minimized is then simply defined as the sum of the mismatch metrics for each constraining tracer field as

$$\hat{f}(\boldsymbol{p}) = f_{[\mathrm{DIP}]} + f_{[\mathrm{DIC}]} + f_{[\mathrm{TA}]} + f_{[\mathrm{O_2}]} + c. \tag{B4}$$

$\hat{f}(\boldsymbol{p})$ is a function of the parameters $\boldsymbol{p}$ because [DIP], [DIC], [TA], and [O$_2$] are taken from the $\boldsymbol{p}$-dependent steady-state solution. $\hat{f}(\boldsymbol{p})$ also includes a small penalty $c$ for the parameters, which prevents unrealistic values. In practice, we use MATLAB's unconstrained minimizer to find an optimal $\boldsymbol{p}$.

For the parameter penalty $c$, we prescribe strict bounds on each optimizable parameter $p$, such that $p$ remains in $(a,b)$. We calculate the penalty as $c = \frac{\omega}{2}\sum_p \hat{p}^2$, where the weight $\omega = 1e-3$ ensures that the parameter-penalty cost is smaller than the tracer cost, and $\hat{p} = \log(\frac{p-a}{b-p})$ transforms $p$ from the interval $(a,b)$ to $\hat{p}$ on the interval$(-\infty,+\infty)$. The penalty for each parameter can be understood as a measure of its negative log-likelihood given a logit-normal distribution on $(a,b)$ as its prior (with mean 0 and standard deviation 1 for its logit). The specified parameter ranges $(a,b)$ are collected in Table B1.

## Appendix C: Key Circulation Characteristics

Figure C1 shows the mixed-layer depths (MLDs) used in OCIM2 and ACCESS-M. For both transport matrices a large vertical diffusivity of $0.1\,\mathrm{m^2\,s^{-1}}$ is used within the mixed layer. The MLD patterns are similar across models except at high latitudes.



**Figure C1.** Annual maximum mixed-layer depth as used in the OCIM2 (a) and ACCESS-M (b) transport matrices.

Most strikingly near the Weddell and Ross Seas the ACCESS-M MLDs reach the sea floor while the observation-based MLDs used in OCIM2 are only a few hundred meters in these regions. The unrealistically deep mixed layer in the Weddell Sea was reported for the 500-yr ACCESS1.3 benchmark run of Bi et al. (2013b), although that run did not have the deep mixed layer in the Ross Sea that was present in the ACCESS1.3 runs on which ACCESS-M is based.





## Appendix D: Biogeochemical Diagnostic Computations

### D1 Export Production

Export production via a given carbon species ($POC_f$, $POC_s$, DOC, or PIC), referred to here as a specific export *pathway*, is calculated using a Green function approach. Below we detail the computation for $POC_f$ as an example; the calculation for the other pathways is similar. We first replace nonlinear processes with equivalent linear terms. For $POC_f$ we have

$$\mathcal{S}_f\left[POC_f\right] = \sigma_f U_C - \mathcal{R}_f\left[POC_f\right] - \mathcal{D}_f\left[POC_f\right], \tag{D1}$$

where the local respiration and dissolution rates of Eq. (3) have been recast in terms of rate coefficients $\mathcal{R}_f$ and $\mathcal{D}_f$ diagnosed from the full nonlinear solution as $\mathcal{R}_f = R_{POC_f}/[POC_f]$ and $\mathcal{D}_f = D_{POC_f}/[POC_f]$. Note that with these coefficients equations (D1) and (3) have the same solution. We may think of $POC_f$ in (D1) as a linear labeling tracer that is attached to the actual $POC_f$ and participates in nonlinear processes in proportion to the $POC_f$ concentration. Linear labeling tracers have been very useful in a number of contexts (e.g., Holzer et al., 2014; Pasquier and Holzer, 2018; Holzer and DeVries, 2022).

Denoting $\mathcal{A}_f = \mathcal{S}_f + \mathcal{R}_f + \mathcal{D}_f$, the Green function $G(\boldsymbol{r},t|\boldsymbol{r}',t')$ that is solution to

$$\partial_t G + \mathcal{A}_f\, G = \delta(t - t')\,\delta(\boldsymbol{r} - \boldsymbol{r}') \tag{D2}$$

gives us the $POC_f$ contribution at location $\boldsymbol{r}$ from carbon uptake at $\boldsymbol{r}'$ through the convolution

$$[POC_f](\boldsymbol{r}|\boldsymbol{r}') = \int\limits_{-\infty}^{t} \mathrm{d}t'\, G(\boldsymbol{r},t|\boldsymbol{r}',t')\,\sigma_f U_C(\boldsymbol{r}'). \tag{D3}$$

In steady-state discretized matrix form, the $\mathrm{d}t'$ integral of the Green function becomes the inverse matrix of the steady-state operator $\mathcal{A}_f$ so that

$$[\mathbf{POC_f}] = \mathbf{A}_f^{-1}\,\sigma_f\,\mathrm{diag}(\boldsymbol{U}_C)\,\mathbf{V}^{-1}, \tag{D4}$$

where $\mathrm{diag}(\boldsymbol{U}_C)$ is a matrix with $\boldsymbol{U}_C$ along its diagonal and where we have multiplied on the right by $\mathbf{V}^{-1}$ to obtain the contribution per unit $\boldsymbol{r}'$ volume.

The global export per unit volume due to production at $\boldsymbol{r}'$ is then obtained by integrating the respiration rate of $[\mathbf{POC_f}]$ over $\boldsymbol{r}$ in the aphotic domain $\Omega_a$:

$$\phi_{POC_f}(\boldsymbol{r}') = \int\limits_{\Omega_a} \mathrm{d}\boldsymbol{r}^3\, \mathcal{R}_f(\boldsymbol{r})\,[POC_f](\boldsymbol{r}|\boldsymbol{r}'). \tag{D5}$$

In matrix form this becomes

$$\phi_{POC_f} = \boldsymbol{\Omega}_a^\mathsf{T}\,\mathbf{V}\,\mathbf{R}_f\,\mathbf{A}_f^{-1}\,\sigma_f\,\mathrm{diag}(\boldsymbol{U}_C)\,\mathbf{V}^{-1}. \tag{D6}$$

For computational efficiency, we take the transpose and calculate

$$\phi_{POC_f}^\mathsf{T} = \sigma_f\,\mathbf{V}^{-1}\,\mathrm{diag}(\boldsymbol{U}_C)\,\mathbf{A}_f^{-\mathsf{T}}\,\mathbf{R}_f\,\mathbf{V}\,\boldsymbol{\Omega}_a. \tag{D7}$$





## D2 Regenerated and Preformed DIP, DIC, and $O_2$

Preformed concentrations are obtained by propagating euphotic concentrations into the aphotic interior without any interior sources or sinks. Preformed concentrations are thus conveniently calculated by solving

$$\mathcal{T}[X_{\text{pre}}] = \Theta(z - z_{\text{eup}}) \left([X] - [X_{\text{pre}}]\right)/\tau_0, \tag{D8}$$

where $X$ can denote DIP, DIC, or $O_2$, $X_{\text{pre}}$ its preformed part, and where the term on the right clamps the preformed concentration to the total concentration in the euphotic zone with rapid time scale $\tau_0 = 1\,\text{s}$. (There is no sensitivity to the exact value of $\tau_0$ as long as it is much smaller than the relevant transport timescales.) In matrix form, Eq. (D8) becomes

$$\boldsymbol{X}_{\text{pre}} = (\mathbf{T} + \mathbf{M}_0)^{-1}\,\mathbf{M}_0\,\boldsymbol{X}, \tag{D9}$$

where $\mathbf{T}$ is the transport matrix, $\boldsymbol{X}$ is the vector of simulated DIP, DIC, or $O_2$ concentrations, and $\mathbf{M}_0$ is a matrix with the mask vector for the euphotic zone divided by $\tau_0$ along the diagonal.

Conversely, regenerated tracers are obtained by labelling them at regeneration and removing them upon entry into the euphotic layer. They can thus be calculated by solving

$$\mathcal{T}[X_{\text{reg}}] = R - \Theta(z - z_{\text{eup}})[X_{\text{reg}}]/\tau_0, \tag{D10}$$

where the regenerated tracer $X_{\text{reg}}$ is clamped to zero in the euphotic zone and $R$ is the corresponding regeneration rate per unit volume (e.g., for $POC_f$-regenerated DIC we use $R = R_{POC_f}$). In matrix form, Eq. (D10) becomes

$$\boldsymbol{X}_{\text{reg}} = (\mathbf{T} + \mathbf{M}_0)^{-1}\,\boldsymbol{R} \tag{D11}$$

where $\boldsymbol{R}$ is the vector of the discretized $R$.

## Appendix E: Sensitivity to BGC Model and Circulation

For a given metric or parameter $X_{b,c}$ that depends on biogeochemical (BGC) model $b$ and circulation $c$, we define its sensitivity to the choice of BGC model as

$$\Delta X_{\text{bgc}} = \frac{1}{n_{\text{circ}}} \sum_c \frac{1}{\overline{X}_c} \sqrt{\frac{1}{n_{\text{bgc}}} \sum_b (X_{b,c} - \overline{X}_c)^2}, \tag{E1}$$

where $n_{\text{circ}} = 2$ is the number of circulations used (OCIM2 and ACCESS-M), $n_{\text{bgc}} = 2$ is the number of BGC models (PC and PCO2), and $\overline{X}_c$ is the mean of $X_{b,c}$ across BGC models at fixed circulation $c$. Sensitivity to circulation $\Delta X_{\text{circ}}$ is given by

710 interchanging $b$ and $c$ in Eq. (E1).

*Author contributions.* BP and MH conceived and designed the study, wrote and ran the code, and created the figures. All authors contributed to writing the paper. Funding was procured by MH, NLB, RJM, and FP.



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
