# Peer review of "Optimal parameters for the ocean's nutrient, carbon, and oxygen cycles compensate for circulation biases but replumb the biological pump"

_EGUsphere, 2023_

## Author Response (AR1)

Below please find our point-by-point responses to the referees that we implemented in the submitted revised manuscript. These responses are essentially copies of the replies that we had previously uploaded during the interactive discussion but updated to match the revised manuscript. In addition, we have corrected a small mistake associated with the numerical values of the PC model (not affecting any conclusions) and we have edited the manuscript throughout for further clarity.

**Response to Referee #1**

We thank Referee #1 for the positive feedback. Our clarifications to the Referee's points are as follows:

**would it make sense to compare the circulations using simpler tracer models, in order to separate the hydrodynamic and biogeochemical problems ? For example, would it be relevant to circulate an age tracer ? The authors seem to have done something in this direction (around lines 165) ? Another comparison could simply concern the vertical velocities or transports accross particular depths between the 2 circulations ? Also, please specify the vertical discretisation of the 2 circulations (z or sigma etc) to ease the reader's experience.**

Yes, an age tracer can be used to disentangle the effects of circulation and biogeochemistry. Detailed analyses of the model circulations, which included ideal mean age (mean time since surface contact) and mean re-exposure time (mean time until next surface contact), have been carried out for ACCESS by Chamberlain et al. (2019) and Holzer et al. (2020), and for OCIM2 by DeVries and Holzer (2019). These references were already cited at various points in the manuscript, but in the revisions, we have now added

> For detailed analyses of the OCIM2 circulation, including ideal mean age (mean time since surface contact) and mean re-exposure time (mean time until next surface contact), see the work by DeVries and Holzer (2019).

and

> For detailed analyses of the circulation captured by the ACCESS1.3 transport matrix, including ideal mean age and mean re-exposure time, see the work by Chamberlain et al. (2019) and Holzer et al. (2020).

where the circulations are first introduced in the Methods section.

(We mention the ideal mean age in the manuscript around line 165 because we used it to quantify the effect of the horizontal coarse graining ($1° \times 1°$ to $2° \times 2°$), which guided our reduction of the vertical mixing to compensate for the increased numerical diffusion.)

Following your suggestion, we have added to the revised manuscript that both matrix models use a vertical depth (as opposed to density) coordinate with layer thicknesses that increase with depth and span the range 10–335 m for ACCESS-M (50 levels) and 36–634 m for OCIM2 (24 levels).

**in the authors' opinion, would the situation be different be similar when using more complex biogeochemical models? Could the authors explain how their approach would be different using forward models ?**

Our main results should apply to more complex biogeochemical models, although our detailed quantitative findings are of course model specific. Optimizing biogeochemical parameters to match observations will compensate for the effects of circulation biases on the biogeochemistry, but the optimal parameters are almost certain to alter the inner workings of the biological pump (with production and export efficiency adjusting to the model's re-exposure times) as compared to our best estimate from the data-assimilated model. This is a general feature of optimizing biological-pump parameters in the presence of circulation biases independent of the specifics of the model and its complexity. In response, we have added the following to the discussion on model complexity:

Instead, optimized model parameters take on values that tend to compensate for shortcomings of the circulation and biogeochemical formulation. This generally changes the inner workings of the biological pump with export production and transfer efficiency adjusting to the circulation model's re-exposure times. While only two model complexities have been examined here (PC and PCO2), our main results should apply to more complex biogeochemical models, although our detailed quantitative findings are of course model specific.

Using a forward model would lead to qualitatively similar results, although a forward model would have the benefit of capturing seasonal and interannual variability. However, optimizing biogeochemical parameters with a forward model is generally prohibitively expensive requiring lengthy model spin-ups for each set of parameters to be explored. By contrast, the transport-matrix approach allows us to solve directly for the steady-state, which is computationally orders or magnitude more efficient than running a forward model to equilibrium (steady-state). In response, we have added the following in the discussion on caveats that include the lack of seasonality:

> Note that using a time-stepped forward model would have the benefit of capturing seasonal and inter-annual variability but would otherwise likely lead to qualitatively similar results at steeply increased computational cost.

**the authors use at different places the word "bias" (e.g. line 280) as opposed to the word "errors". I suppose the circulation is indeed biased; can the authors explain whether this is systematic ? Can the present study be used as valuable information for future version of the circulations ?**

We use the term "bias" deliberately because the ACCESS ocean circulation errors are systematic and characterized by coherent patterns as has been reported by Bi et al. (2013) (who also use the language of "biases"). We do hope that our work will be useful for future versions of the ACCESS ocean model, which can then be assessed not just in terms of the fidelity of the physical variables but also in terms of key biogeochemical metrics. In response, we have added the following at the end of the revised manuscript inviting other models to be similarly benchmarked using diagnostics of the biological pump:

> We hope that our work will lead to future ocean models being assessed not just in terms of the fidelity of their physical variables, but also in terms of key biogeochemical metrics.

**As the authors focus on the 1990's, would it be possible to couple PCO2 (or PC) to the Copernicus Marine (CMEMS) GLObal model product (1993-present) ?**

It is certainly possible to embed the PC or PCO2 biogeochemistry model into any other circulation, including CMEMS-based circulations. However, this is beyond the scope of our study and would require significant work. Specifically, one would at a minimum need to (i) extract a transport matrix as was done for the ACCESS model from a relevant output, (ii) significantly coarsen in the horizontal to make the model numerically affordable or, alternatively, use different solvers suitable for much larger nonlinear systems (e.g., using matrix-free Newton-Krylov), (iii) determine appropriate eddy diffusion schemes, and (iv) carefully assess the resulting deep circulation (e.g., as quantified by ventilation tracers) to ensure fidelity to the parent model. No change to the manuscript in response.

**Response to Referee #2**

We thank the referee for the positive feedback. Our point-by-point response follows below:

**(1) Eqn. 3 and lines 84-85: left hand term $\mathcal{S}_{\mathrm{f}}[\mathrm{POP_f}]$ and $\mathcal{S}_{\mathrm{s}}[\mathrm{POP_s}]$ and "is the divergence of the flux of the $\mathrm{POP}_k$ tracer with sinking speed $w_k$" — Does this mean that the POP tracers are not subject to advection and diffusion (as for DOP and DIP)? If so (if particulate and dissolved tracers are treated differently) would this in any way affect the solution/evaluation of the Green function (appendix D1 and D2)?**

Yes, the particles are not advected and diffused along with the water. This approximation is justified because the transport of particles is generally dominated by their sinking. The approximation is desirable because adding the full advection–diffusion operator for the particles would reduce the sparsity of the system of equations, making it computationally more expensive to solve. This does not affect the Green-function approach of our diagnostics in any way. In the revisions, we have added the following sentence:

> Note that here particles are only subjected to gravitational sinking; including their advective–diffusive transport does not significantly change the solutions but greatly increases computational cost.

**(2) Eqn. 3 and lines 88-89: "The global phosphate inventory is prescribed by weakly restoring DIP to the global observed mean [DIP]" - Why is restoring necessary? Is there any loss or gain of phosphate in the model? In particular: what happens to sinking POP at the lower model boundary (given the " unrealistic POM accumulation in the bottom grid boxes under anoxic conditions" mentioned in line 100)? Is there any loss to the sediment, that necessitates the restoring? How big are the restoring terms for DIP (and TA)?**

The total amount of phosphate is conserved in our model. Because we are directly solving for the steady state, we cannot specify an initial phosphate inventory. Without the restoring term, the steady-state phosphate equation would be singular (rank-deficient). The weak ("geological") restoring removes this deficiency (null space) from the system. (This has been discussed in several previous publications by the authors, e.g., Kwon and Primeau (2006); Primeau et al. (2013); Holzer et al. (2021).) POP that reaches the bottom grid box superjacent to the seafloor accumulates there (no loss to the sediments) until it is eventually remineralized to DIP or solubilized to DOP, both of which are then transported with water and mixed back into surrounding grid boxes. The DIP restoring term is multiple orders of magnitude smaller than all other DIP tendencies. In response, we have added the following:

> The global phosphate inventory is constant in our model and prescribed by weakly restoring DIP to the  observed global mean $\overline{[\mathrm{DIP}]} = 2.17\,\mu\mathrm{M}$ via $J_{\mathrm{DIP}}^{\mathrm{geo}} = (\overline{[\mathrm{DIP}]} - [\mathrm{DIP}])/\tau_{\mathrm{geo}}$ with "geological" timescale $\tau_{\mathrm{geo}} = 1\,\mathrm{Myr}$. Without prescribing the total amount of phosphate in this way there would be no unique solution to steady-state Eq. (1). (This contrasts with time-stepped models where, in the absence of external sources and sinks, the total amount of phosphate is set by the initial conditions.)

**(3) Lines 98-100: "Note, however, that Eq. (2) differs from previous parameterizations in that we implicitly include the effect of microbes switching to nitrate for organic matter oxidization (denitrification) by disallowing respiration rates to decline in anoxic waters below $[\mathrm{O_2^{lim}}] = 5\,\mu\mathrm{M}$." - Does this mean respiration continues in the absence of oxygen? If so, is the oxygen dept accounted for? Would it play a large role, if it were accounted for in the cost function?**

Yes, this means that respiration continues when $[\mathrm{O_2}]$ falls below $[\mathrm{O_2^{lim}}]$ at the same rate as would occur if $[\mathrm{O_2}]$ was equal to $[\mathrm{O_2^{lim}}]$, but without utilizing $[\mathrm{O_2}]$. The effect of this on oxygen deficit (as measured by TOU for example) is consistently accounted for by the oxygen equation without the need for extra modelling because we optimize the mismatch with $[\mathrm{O_2}]$ not oxygen deficit. We experimented with various other parameterizations of denitrification and found the solutions to be very similar. In response, we have added the following after Eq. (2):

> where $T_{\mathrm{ref}} = 20\,°\mathrm{C}$  and $[\mathrm{O_2^{lim}}] = 5\,\mu\mathrm{M}$ defines the oxygen limit below which water is deemed anoxic. Note that Eq. (2) differs from previous parameterizations in that we  include the effect of microbes switching to nitrate for  organic-matter oxidization (denitrification) by explicitly disallowing respiration rates to decline in anoxic waters  through the max function in Eq. (2). This

means that respiration continues when $[O_2]$ falls below $[O_2^{lim}]$ at the same per-molecule rate as would occur if $[O_2]$ was equal to $[O_2^{lim}]$ but without utilizing oxygen. (Note that $[O_2]$ can fall below $[O_2^{lim}]$ in spite of this being explicitly disallowed in the oxygen tracer equation (5) discussed below because we smooth of step functions for differentiability as described in Appendix B3.)

**(4) In the model POP may sink very deeply, and dissolves to DOP. Is DOP remineralisation (to DIP) also oxygen dependent as for POP? Does DOP also accumulate at the bottom or elsewhere? Could it be that any improvement in PO4 (through optimisation of sigma) comes at the cost of DOP mismatches (e.g., Kriest, 2017, https://doi.org/10.5194/bg-14-4965-2017)?**

For simplicity, DOP is remineralized with a fixed 2-yr timescale at rate $D_{POM} = [POM]/\tau_{POM}$ that is independent of $[O_2]$ (see also lines 101–103). DOP does not accumulate on the seafloor, but is transported with seawater. Only POP is allowed to accumulate at the seafloor, in which case the accumulated POP is eventually remineralized to DIP or dissolved into DOP, both of which are transported with water into the neighbouring boxes. It is entirely that improvements in the DIP mismatch come at the cost of DOP realism. However, we deliberately do not use DOP observations as a constraint on our model because our representation of DOP with a single semi-labile pool is too simplistic to be properly constrained by observations. No changes to the manuscript.

**(5) Optimisation: Could you briefly explain what is behind "MATLAB's unconstrained minimizer" (appendix B4, lines 646-647)? Is this a gradient-based method? If so, is there the possibility that the optimisation especially of ACCESS became trapped in a local minimum?**

We used the MATLAB routine `fminunc`, which is indeed gradient-based. While it is theoretically possible that the optimization is trapped in a local minimum, during this research we performed many optimizations for slightly different versions of the model and they generally approached a similar state. This passage in the appendix has been extended and now reads

> In practice, we use MATLAB's unconstrained minimizer, `fminunc`, to find an optimal $\boldsymbol{p}$. We note in passing that the minimum of the objective function $\hat{f}$ determined in this way is not guaranteed to be the global minimum given the complex nature of $\hat{f}$. However, during the course of this research, we optimized many versions of our biogeochemical model and found that they all converged to a similar minimum.

**(6) Preformed oxygen simulated by the models is mentioned several times (lines 220, 228, 307) - For me it would be very interesting and helpful to see plots (e.g., zonal means) of this and/or regenerated nutrients, especially to understand the following results on different pathways in the model better.**

We now provide plots of basin zonal-mean preformed and regenerated DIP and preformed $[O_2]$ and TOU in a supplementary file (not part of the Appendices).

**(7) Lines 239ff "When O2 is prescribed from observations (PC model) ..." - This is one of few references to the PC model, which later serves as an alternative model. Are there any differences in the spatial distribution (zonal means) of DIP? A few words on this (or a plot of zonal means in the supplement) could be helpful, as this model later (Table 1) serves as an alternative biogeochemistry.**

There are minor differences in the zonal means of DIP ($\pm 0.1\,\mu M$) between the PCO2 and PC models. In response, we now provide plots of the basin zonal means of DIP in a supplementary file.

**(8) Lines 269ff: Cross-validation experiments are mentioned (simulating ACCESS with OCIM-optimal parameters). As the two different circulations seem to require very different parameters (namely those that affect remineralisation), I think it would be very interesting to see zonal plots of DIP and oxygen for the cross-validation experiments.**

In a supplementary file, we now also provide basin zonal means of the ACCESS-M tracers (DIP, DIC, $O_2$) obtained with OCIM2 parameters, and their difference with the optimized ACCESS-M tracers.

**Lines 221ff: "Instead, the Southern Ocean POC respiration rate is weaker for ACCESS-M, allowing more oxygen to be mixed throughout the water column. (The reduced respiration rate is largely driven by a lower optimal value of $\gamma$s (Table 1), with POCs dominating respiration for ACCESS-M as further discussed in subsection 3.3.2.)" - Why is the respiration rate weaker in ACCESS-M? From what I see in Table 1, (basic) optimal respiration rates are 1.05 and 0.535 for large and small POP, and thus much larger than those of OCIM? Likewise, the Q10 values of ACCESS-M are larger. Is this because of lower temperatures in the Southern Ocean in ACCESS-M?**

We thank the referee for catching this mistake. We have revised the paragraph to state:

> Instead, the  unrealistically deep vertical mixing dramatically reduces $O_2$ residence times for ACCESS-M  such that total oxygen is closer to preformed oxygen in the Southern Ocean than is the case for OCIM2. This occurs despite the larger ACCESS-M  respiration-rate coefficients ($\gamma_s$ and $\gamma_f$) and lower $q_{10}$ (Table 1) presumably because of the relatively low organic-matter production in the polar Southern Ocean (Fig. 4 below).

**Eqn 2: Is temperature given in K or °C? (because of "K" in the denominator of the exponent)**

This equation involves the temperature difference $T - T_{\mathrm{ref}}$, for which it does not matter if the unit is °C or K. Having the denominator as $10\,\mathrm{K}$ has the advantage of being unambiguous — if $10\,°\mathrm{C}$ was used, one might misinterpret the denominator as $(10 + 273.15)\,\mathrm{K}$, which would be incorrect. We think the equation is clear as written (no changes to the manuscript).

**Eqn A1 and line 578: The same PAR (light) was used for both models - does this relate only to the surface light, or also the light at depth?? If the latter, then the approach neglects self-shading by phytoplankton, and hence any feedback effects of phytoplankton on production, correct? This might be an important difference to more complex BGC models, together with the assumed constant phytoplankton concentration.**

Yes, we prescribe the same approximately exponentially attenuating PAR throughout for both models, taken from the same ACCESS1.3 run used to build the ACCESS-M matrix. Hence, the effects of self-shading are not modelled for simplicity. We agree that a more complex forward model could account for this. We have revised the manuscript to now say:

> PAR is prescribed throughout the water column from the ACCESS1.3 model runs  for both the ACCESS-M- and OCIM2-embedded PCO2 models. (PAR is therefore not coupled to the plankton concentration, precluding any effects from self-shading.)

**Eqn D1: Is there a factor (1-sigma) missing in this equation (as in Eqn 3)? Would this also affect the following equations (D4, D6, D7)?**

We thank the Referee for catching this typo. A factor $(1-\sigma)$ was indeed missing. This has been corrected in the revised version.

**Table B1 (and Table 1): It would be very helpful to have the meaning of the parameters (e.g., "fraction of production routed to DOP" for sigma) in at least one of the tables.**

We agree and have added a descriptive column to Table 1 in the revised manuscript.

**References**

Bi, D., Marsland, S. J., Uotila, P., O'Farrell, S., Fiedler, R. A. S., Sullivan, A., Griffies, S. M., Zhou, X., and Hirst, A. C.: ACCESS-OM: the Ocean and Sea ice Core of the ACCESS Coupled Model, Australian Meteorological and Oceanographic Journal, 63, 213–232, 2013.

Chamberlain, M. A., Matear, R. J., Holzer, M., Bi, D., and Marsland, S. J.: Transport matrices from standard ocean-model output and quantifying circulation response to climate change, Ocean Modelling, 135, 1–13, https://doi.org/10.1016/j.ocemod.2019.01.005, 2019.

DeVries, T. and Holzer, M.: Radiocarbon and Helium Isotope Constraints on Deep Ocean Ventilation and Mantle-[3]He Sources, Journal of Geophysical Research: Oceans, 124, 3036–3057, https://doi.org/10.1029/2018JC014716, 2019.

Holzer, M., Chamberlain, M. A., and Matear, R. J.: Climate-Driven Changes in the Ocean's Ventilation Pathways and Time Scales Diagnosed From Transport Matrices, Journal of Geophysical Research: Oceans, 125, e2020JC016 414, https://doi.org/10.1029/2020JC016414, e2020JC016414 10.1029/2020JC016414, 2020.

Holzer, M., Kwon, E. Y., and Pasquier, B.: A New Metric of the Biological Carbon Pump: Number of Pump Passages and Its Control on Atmospheric $pCO_2$, Global Biogeochemical Cycles, 35, e2020GB006 863, https://doi.org/10.1029/2020GB006863, e2020GB006863 2020GB006863, 2021.

Kwon, E. Y. and Primeau, F. W.: Optimization and sensitivity study of a biogeochemistry ocean model using an implicit solver and in situ phosphate data, Global Biogeochemical Cycles, 20, https://doi.org/10.1029/2005GB002631, 2006.

Primeau, F. W., Holzer, M., and DeVries, T.: Southern Ocean nutrient trapping and the efficiency of the biological pump, Journal of Geophysical Research: Oceans, 118, 2547–2564, https://doi.org/10.1002/jgrc.20181, 2013.